# Murine Fam65b forms ring-like structures at the base of stereocilia critical for mechanosensory hair cell function

Bo Zhao[1,2]*, Zizhen Wu[1,2], Ulrich Müller[1,2]*

[1]Department of Molecular and Cellular Neuroscience, The Scripps Research Institute, La Jolla, United States; [2]Dorris Neuroscience Center, The Scripps Research Institute, La Jolla, United States

**Abstract** Cochlear hair cells convert sound-induced vibration into electrical signals. *FAM65B* mutations cause hearing loss by an unknown mechanism. Using biochemistry and stochastic optical reconstruction microscopy (STORM), we show here that Fam65b oligomers form a circumferential ring near the basal taper of the mechanically sensitive stereocilia of murine hair cells. Taperin, a second protein near the taper, forms a dense-core-like structure that is disrupted in the absence of Fam65b. Stereocilia of *Fam65b*-deficient murine hair cells start to develop, but mechanotransduction is affected and stereocilia deteriorate. Yeast-two-hybrid screens identify RhoC as a Fam65b binding partner. RhoC co-localizes with Fam65b in stereocilia and regulates Fam65b oligomerization. Binding to RhoC and oligomerization are critical for Fam65b function. Our findings thus reveal a highly organized compartment near the base of stereocilia that is critical for hair cell function and affected in disease.

*For correspondence: zhaobo@scripps.edu (BZ); umueller@scripps.edu (UM)

## Introduction

Hair cells of the mammalian cochlea have a specialized morphology that is optimized for their role as mechanical sensors for sound. Each hair cell contains at the apical surface a bundle of stereocilia that are organized in rows of decreasing height (*Petit and Richardson, 2009*; *Schwander et al., 2010*). Stereocilia are cylindrical in shape but form at their base a taper (*Figure 1*) that is critical for their function because stereocilia pivot at their base during mechanical stimulation (*Fettiplace and Kim, 2014*; *Müller and Barr-Gillespie, 2015*; *Zhao and Müller, 2015*). Confocal imaging of isolated bullfrog hair cells has shown that the membrane of stereocilia is spatially segregated into at least three domains including a specialized basal taper domain (*Zhao et al., 2012*). Genetic studies combined with immunolocalization experiments have identified several proteins that are concentrated at or near the taper. Cell surface receptors such as PTPRQ, USH2A and VLGR1 form extracellular filaments near the taper and recruit several cytoplasmic proteins that might link the receptors to the cytoskeleton (*Adato et al., 2005*; *McGee et al., 2006*; *Michalski et al., 2007*; *Salles et al., 2014*). Myosin motor proteins such as Myo6 and Myo7A are also localized near the taper (*Hasson et al., 1997*) and may be important for protein transport or for adjusting tension between the cell membrane and the cytoskeleton. Proteins with potential roles in regulating F-actin dynamics and/or anchorage to the cell membrane such as radixin and taperin are concentrated at the taper region as well (*Pataky et al., 2004*; *Rehman et al., 2010*; *Salles et al., 2014*). In addition, TRIOBP forms rootlets that extend from stereocilia into the cuticular plate and traverse the taper region (*Kitajiri et al. 2010*). Importantly, mutations in the genes that encode proteins localized near the taper cause inherited forms of deafness (*Ahmed et al., 2003*; *Khan et al., 2007*; *Liu et al., 1997*; *Rehman et al., 2010*; *Schraders et al., 2010*; *Weil et al., 1997*), thus reinforcing the concept that

the taper is a specialized and functionally important subdomain within stereocilia. However, the overall organization of proteins at the base of stereocilia and the mechanisms that regulate their assembly and function are not known.

Recent studies have shown that a mutation in *FAM65B* causes hearing loss in humans (*Diaz-Horta et al., 2014*). Immunolocalization experiments have provided evidence that Fam65b is expressed in hair cells and knock-down of *Fam65b* in zebrafish leads to the loss of hair cells (*Diaz-Horta et al., 2014*). However, the mechanisms by which Fam65b affects hair cell function are unclear. In some cell types, Fam65b appears to be localized at the cell membrane (*Diaz-Horta et al., 2014*), but a direct link of Fam65b to the cytoskeleton has also been proposed (*Yoon et al., 2007*). Accordingly, overexpression of Fam65b in C2C12 myoblasts and in HEK293 kidney cells leads to the formation of filopodia (*Yoon et al., 2007*), while studies in neutrophils and T-lymphocytes suggest that Fam65b can regulate RhoA activity (*Gao et al., 2015*; *Rougerie et al., 2013*).

To further define the function of Fam65b in mechanosensory hair cells, we have studied its subcellular distribution in hair cells by stochastic optical reconstruction microscopy (STORM) and generated *Fam65b*-deficient mice. We show here that Fam65b is localized near the basal taper of hair-cell stereocilia where it forms oligomers by head-to-head and tail-to-tail interaction to form a circumferential ring that does not penetrate the stereociliary core. In contrast, taperin is localized throughout the stereociliary core. Biochemical data suggest that Fam65b does not bind to taperin or any of the proteins that are known to bind to taperin. Instead, using a yeast-two-hybrid screen, we have identified RhoC as a binding partner for Fam65b and show that RhoC regulates Fam65b oligomerization. Defects in Fam65b expression, oligomerization and binding to RhoC affect stereocilia development and function and cause deafness. Our findings thus reveal a striking spatial segregation of proteins within the basal domain of stereocilia, provide insights into the mechanisms that regulate protein assembly in this critical functional domain and reveal a role of Fam65b for normal stereocilia morphogenesis and function.

## Results

### *Fam65b*-deficient mice are deaf and show defects in hair cell function

In order to gain insights into the mechanisms by which mutations in *Fam65b* cause hearing impairment, we created a *Fam65b*-deficient mouse line. In the mutant mice, the coding region of the *Fam65b* gene was replaced with a *LacZ* transgene. We will refer to the modified allele as *Fam65b$^{LacZ}$* (*Figure 1B*). Next we generated homozygous *Fam65b$^{LacZ/LacZ}$* mice. Measurements of the auditory brain stem response (ABR) to broadband click stimuli in 4-week old animals revealed that homozygous mutant mice were deaf (*Figure 1C,D*). Recordings of ABRs in response to pure tones revealed that *Fam65b$^{LacZ/LacZ}$* mice were deaf across the entire analyzed frequency spectrum (*Figure 1E*). Hearing function was minimally affected in heterozygous *Fam65b$^{LacZ/+}$* mice (*Figure 1E*), consistent with a recessive mode of inheritance of the auditory phenotype.

We next recorded distortion product otoacoustic emissions (DPOAEs), which are mechanical distortions generated in the inner ear when two primary tones ($f_1$ and $f_2$) are presented. Outer hair cells (OHCs) amplify the distortions and they are propagated back through the middle ear and ear canal and can be measured in sound pressure waveforms (*Kemp, 1978*; *Shaffer et al., 2003*). DPOAEs were absent in *Fam65b$^{LacZ/LacZ}$* mice at 4 weeks of age at all frequencies tested (*Figure 1F,G*). As these emissions depend on the mechanical activity of OHCs, we conclude that OHC function was affected in *Fam65b$^{LacZ/LacZ}$* mice.

### Fam65b expression in hair cells of the inner ear

To gain insights into the mechanism by which mutations in *Fam65b* affect hearing function, we next analyzed its expression pattern in the inner ear. Taking advantage of the *LacZ* insertion within the genomic locus of the *Fam65b* gene, we analyzed in heterozygous *Fam65b$^{LacZ/+}$* mice the expression of the *Fam65b* gene by X-gal staining of cochlear whole mounts at postnatal day 4 (P4) (*Figure 2A–C*). The *Fam65b* gene was expressed along the entire length of the cochlear duct (*Figure 2A*) with strongest expression in inner hair cells (IHCs), OHCs and Hensen's cells (*Figure 2B,C*). During cloning of the full-length *Fam65b* cDNA from an inner ear cDNA library, we identified a new *Fam65b*

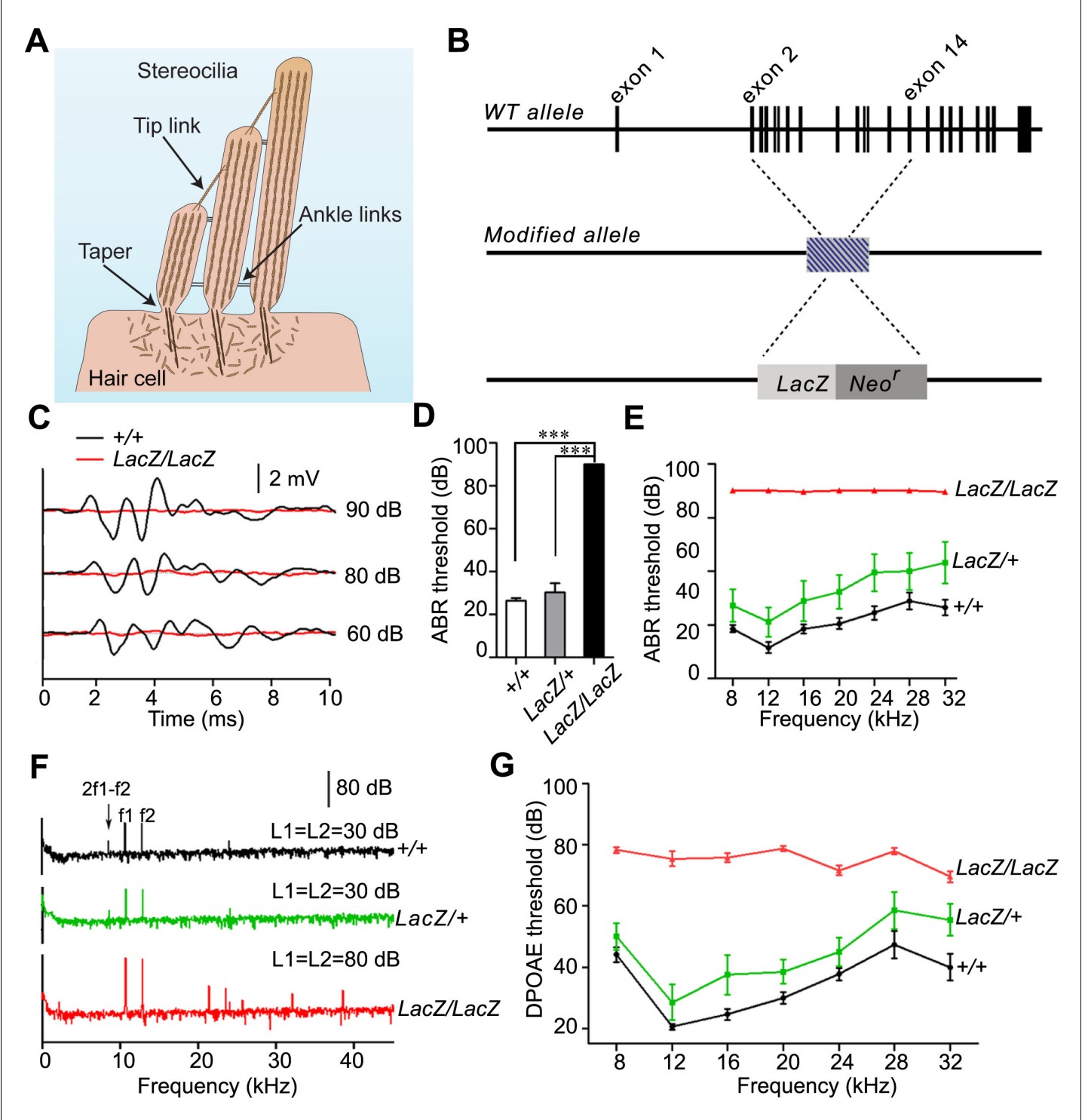

**Figure 1.** Analysis of hearing function in *Fam65b*-deficient mice. (**A**) Diagram of cochlear hair cells showing the mechanically sensitive hair bundle. Stereocilia are arranged into three rows of decreasing heights. Stereocilia form a taper at their base near the insertion site into the apical hair cell surface. (**B**) Diagram of the strategy to generate *Fam65b* knock–out mice. Exons of *Fam65b* were substituted with a *LacZ* expression cassette. (**C**) Representative ABR traces to click stimuli in the indicated control and mutant mice at 4 weeks of age. (**D**) Statistic results of ABR thresholds to click stimuli at 4 weeks of age. Results are represented as mean ± SE (standard error of the mean). ***p<0.001, by Student's t-test. (**E**) ABR thresholds to pure tones at 4 weeks of age. Results are represented as mean ± SE. ***p<0.001 (ANOVA) (**F**) Representative DPOAE response spectra from control and mutant mice at a single stimulus condition (median primary frequency = 12 kHz). Note the 2f1-f2 peak (black arrow), which is absent in mutant

*Figure 1 continued on next page*

*Figure 1 continued*

mice. (**G**) DPOAE thresholds at different frequencies in animals at 4 weeks of age. Results are represent as mean ± SE. ***p<0.001 (ANOVA). More than five animals in each group were tested.

splice isoform that lacks amino acids encoded by exon13. We detected by RT-PCR expression of the smaller *Fam65b* isoform in many tissues including the cochlea, while expression of the larger isoform was confined to the brain including the cerebellum, spinal cord and retina (*Figure 2D*).

Next we analyzed the expression pattern of Fam65b in hair cells by immuno-histochemistry using a commercially available antibody (Sigma). In whole mounts of the cochlea at P5, we observed expression in OHCs and IHCs where Fam65b immunoreactivity outlined the shape of the stereociliary bundle (*Figure 2E*). Additional staining was observed in the microvilli on the apical surface of Hensen's cells (*Figure 2E*), which was consistent with the X-gal staining results (*Figure 2A–C*). We also observed staining in the microvilli of other support cells that did not stain with X-gal. The cells were presumably X-gal negative because expression levels were low and thus difficult to detect with X-gal staining. Clearly, the antibody staining signals were specific because they were not observed in *Fam65b^{LacZ/LacZ}* mice (*Figure 2E*). Higher resolution images showed that Fam65b was expressed in IHCs and OHCs at the basal end of stereocilia near the taper region (*Figure 2F*). To confirm this finding, we used injectoporation (*Xiong et al., 2012*) to express HA-tagged Fam65b in mechanosensory hair cells at P3. We fixed and stained the injectoporated hair cells two days later with phalloidin to reveal stereocilia and with an antibody to HA to detect HA-Fam65b. Consistent with the immuno-localization data using Fam65b antibodies, HA-Fam65b was concentrated near the base of stereocilia (*Figure 2G*). When we expressed by injectoporation a HA-Fam65b construct carrying a deletion linked to deafness in humans (Δ58–111), (*Diaz-Horta et al., 2014*), the protein was distributed diffusely throughout the cytoplasm and no longer targeted to the base of stereocilia (*Figure 2H*).

Previous studies have shown that taperin is localized near the basal taper of stereocilia (*Rehman et al., 2010*). To further define the localization of Fam65b, we compared in histological sections the expression pattern of Fam65b with the expression pattern of taperin. Both proteins showed a similar distribution near the basal taper of stereocilia (*Figure 2I,J*). Finally, to examine the colocalization of Fam65b and taperin, we again expressed by injectoporation HA-Fam65b in mechanosensory hair cells. Co-staining of the hair cells with an antibody to taperin and an antibody to HA confirmed that Fam65b and taperin were co-localized near the taper of stereocilia (*Figure 2K*).

## *Fam65b*-deficiency affects hair bundle morphogenesis and stereociliary growth

The expression pattern of Fam65b suggested that it might regulate stereocilia development, function and/or maintenance. We therefore compared the development of the sensory epithelium in the cochlea of wild-type and *Fam65b^{LacZ/LacZ}* mice in whole mount cochlea using phalloidin staining combined with fluorescence deconvolution microscopy. Whole mount staining at P2 revealed that the sensory epithelium in mutant mice was patterned properly into three rows of OHCs and one row of IHCs (*Figure 3A*). However, hair bundle morphology appeared abnormal in the mutants with aberrant bundle shape relative to controls (*Figure 3A*). Analysis by scanning electron microscopy (SEM) confirmed that hair bundles in the mutants contained a single kinocilium as well as a bundle of stereocilia but the bundles were abnormally shaped (*Figure 3B*). At P2 the bundles of most OHCs appeared similar in size to wild-type and formed a staircase pattern but some OHCs showed defects in hair bundle polarity (*Figure 3A,B*). In contrast, hair bundles of most IHCs were already severely disorganized (*Figure 3A,B*). To quantify polarity defects in OHCs, we determined the deviation of the position of the bundle center, which contains the longest stereocilia, from their normal position at the tip of the V-shaped bundle (*Figure 3E*). Unlike in control heterozygous *Fam65b^{LacZ/+}* mice, bundle orientation of OHCs was far more variable in *Fam65b^{LacZ/LacZ}* mice (*Figure 3F*).

By P5 the bundles of both OHCs and IHCs were severely disorganized and many stereocilia appeared abnormally thick and large indicating that their elongation was affected (*Figure 3C,D*). Interestingly, similar defects in stereociliary growth were observed in vitro. When we prepared cochlear explants from *Fam65b^{LacZ/+}* and *Fam65b^{LacZ/LacZ}* mice at P2 and cultured them for 2 days,

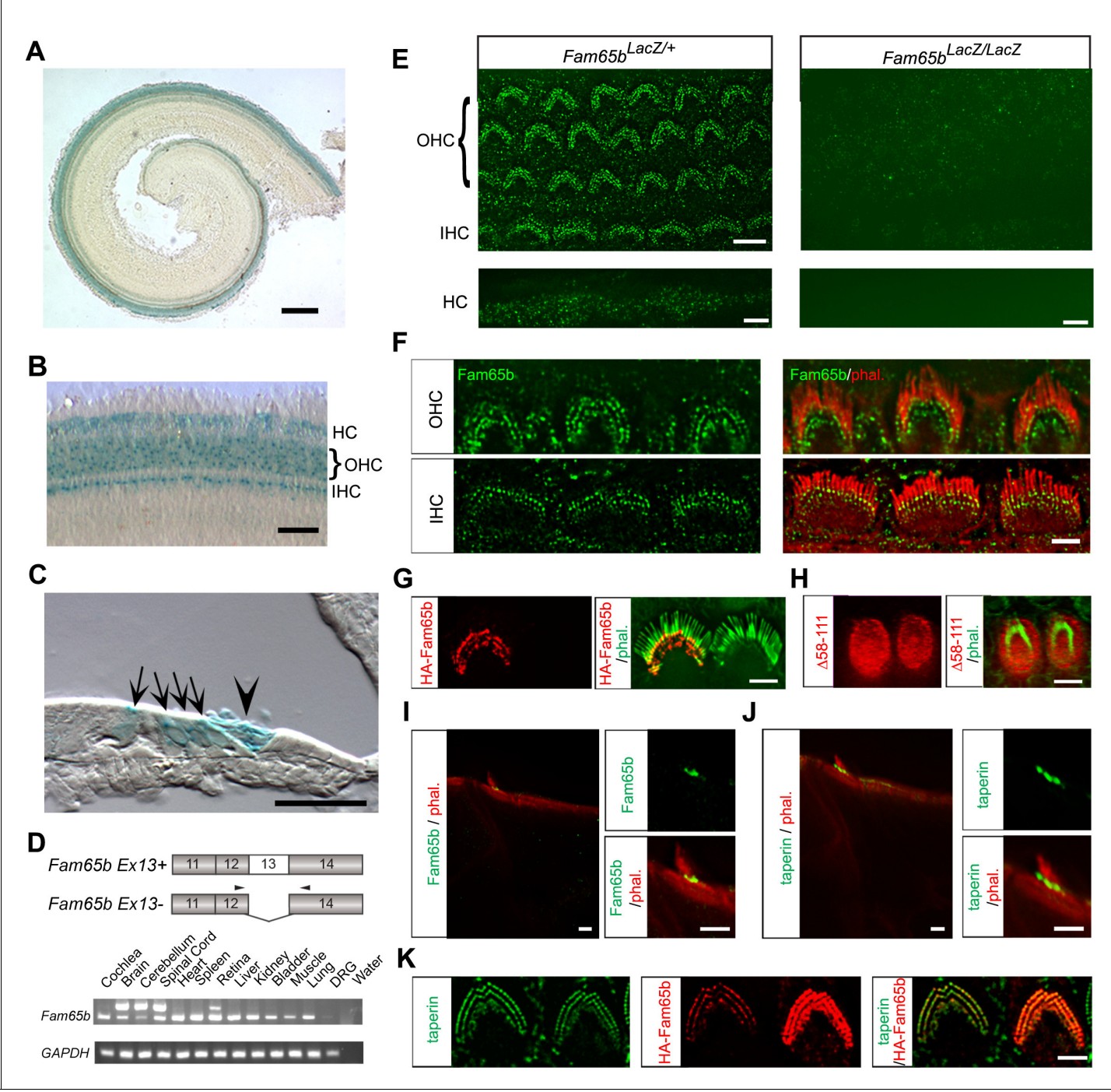

**Figure 2.** Expression of Fam65b in hair cells. (**A–C**) Cochlear whole mounts from P4 *Fam65b*<sup>LacZ/+</sup> mice were stained for LacZ. (**A**) Whole mount staining reveals expression of *Fam65b* along the length of the cochlear duct. (**B**) Higher magnification view of whole mount cochlea. OHCs, IHCs and Hensen's cells (HCs) express LacZ. (**C**) Section through a whole mount revealing LacZ expression in OHCs, IHCs, and HCs. Arrows point to OHCs and, IHCs, arrowhead points to HCs. (**D**) PCR analysis of *Fam65b* isoform expression in different tissues. Upper panel shows the mouse *Fam65b* gene structure. Boxes with numbers represent exons. Arrowheads show positions of primers. Lower panel shows expression of *Fam65b* isoforms in different tissues. GAPDH served as a loading control. Water lane is the negative control. (**E**) Cochlear whole mounts from *Fam65b*<sup>LacZ/+</sup> and *Fam65b*<sup>LacZ/LacZ</sup> mice at P5 were stained for a commercial antibody to Fam65b (Sigma, green) and imaged with fluorescent deconvolution microscopy. The lower panel shows Hensen's cells (HC). Note the absence of a signal in the mutant mice. (**F**) Co-staining of cochlear whole mounts with phalloidin-rhodamine to reveal stereocilia (red) and antibodies to Fam65b (green). Note the localization of Fam65b at the base of stereocilia. (**G**) Cochlear explants were prepared at P3 and injectoporated to express HA-Fam65b. Two days later, the cells were stained with HA-antibody. Note the expression of HA-Fam65b at the base of stereocilia. (**H**) Cochlear explants were prepared at P3 and injectoporated to express HA-Fam65b (Δ58–111), corresponding to a mutation that causes

*Figure 2 continued on next page*

*Figure 2 continued*

deafness in humans (*Diaz-Horta et al., 2014*). Note the diffuse expression of the mutated protein within the cytoplasm of hair cells with no specific localization at basal regions of hair cells. (I, J) Inner ear sections (P5) were stained with Fam65b (I) and taperin (J) antibodies. Tissues were counterstained with phalloidin. Note expression of Fam65b and taperin at the base of stereocilia. (K) Cochlear explants were prepared at P3 and injectoporated to express HA-Fam65b. Two days later, the cells were stained with antibodies against HA (red) and taperin (green). Note colocalization of HA-Fam65b and taperin in injectoporated hair cells. Note that the cell to the right strongly expressed the transgene, while the cell to the left expressed it weakly. Scale bars: (A) 500 μm; (B, C) 50 μm; (E–K) 4 μm.

most of the hair bundles of IHCs from mutant animals were disorganized and grew abnormally long (*Figure 3G*) indicating that bundle cohesion and growth were affected.

To analyze structural defects in hair bundles in more detail, we carried out additional SEM analysis with hair cells at P5. Low magnification views revealed that hair bundles were disorganized in the mutant animals (*Figure 4A*). Morphological defects were variable but we could group them into several classes (*Figure 4B,C*). For OHCs: (I) rounded bundle-shape that deviated from the classical V-shape; (II) elongated but coherent bundles; (III) rounded bundles with signs of degeneration; (IV) fragmented bundles with some thin and elongated stereocilia. For IHCs: (I) coherent bundles with slight abnormalities; (II) coherent bundles with degenerative changes; (IIIa) fragmented bundles with thin and elongated stereocilia; (IIIb) fragmented bundles with few remaining thin and elongated stereocilia. The differences in bundle morphology are indicative of various stages during a degenerative process that is initiated already at early postnatal ages when hair bundles have not reached their mature shape. Quantification revealed that for OHCs class II bundle-morphology was most prevalent at this age, while for IHCs class II and IIIa/b bundles were the most prominent (*Figure 4C*).

The localization pattern of Fam65b in hair cells prompted us to investigate structural changes near the base of stereocilia in more detail. Unlike in the stereocilia in wild-type mice (*Figure 4D,E*), the basal region of stereocilia in mutants was narrow and elongated (*Figure 4F–J*). We also observed in the mutants protrusions from the apical surface of hair cells by SEM (*Figure 4K,L*) and transmission electron microscopy (TEM) (*Figure 4M*) indicative of structural changes in the apical hair cell surface. Despite these strucutal abnormalities, analysis of sensory epithelia in mice at P16 and P28 revealed that hair cells with abnormal hair bundles were maintained with little evidence for hair cell loss (*Figure 4—figure supplement 1*).

Taken together, these findings suggest that perturbations in the basal compartment of stereocilia are likely responsible for the defects in bundle cohesion and growth in *Fam65b*$^{LacZ/LacZ}$ mice. However, hair cells with abnormally shaped bundles are maintained for at least 4 weeks after birth.

## Taperin localization and functional perturbation using shRNA

Our immunolocalization data showed that Fam65b and taperin were similarly expressed at the base of sterocilia. Interestingly, taperin was no longer localized to the base of stereocilia in IHCs from *Fam65b*$^{LacZ/LacZ}$ mice, while the normal distribution of espin near the tips of stereocilia (*Zheng et al., 2014*) was maintained (*Figure 5A,B*). These findings provide additional support for the conclusion that in *Fam65b*$^{LacZ/LacZ}$ mice the integrity and structural organization of the base of stereocilia is affected.

LacZ staining in *Fam65b*$^{LacZ/+}$ mice detected *Fam65b* expression not only in hair cells but also in Hensen's cells. We did observe by SEM no structural defects at the apical surface of Hensen's cells (*Figure 4A* and data not shown). However, to further confirm that the morphological defects of hair cells were cell-autonomous, we generated an shRNA targeting Fam65b, which knocked down Fam65b expression in HEK293 cells in a dose-dependent manner (*Figure 5C*). Using injectoporation, we expressed GFP together with the shRNA targeting Fam65b in P2 OHCs. We identified the injectoporated cells three days later by GFP fluorescence and analyzed hair bundle morphology. Expression of the shRNA targeting Fam65b but not of a scrambled control shRNA severely affected hair bundle morphology (*Figure 5D*). We thus conclude that Fam65b acts at least in part cell autonomously in hair cells to regulate hair bundle morphogenesis.

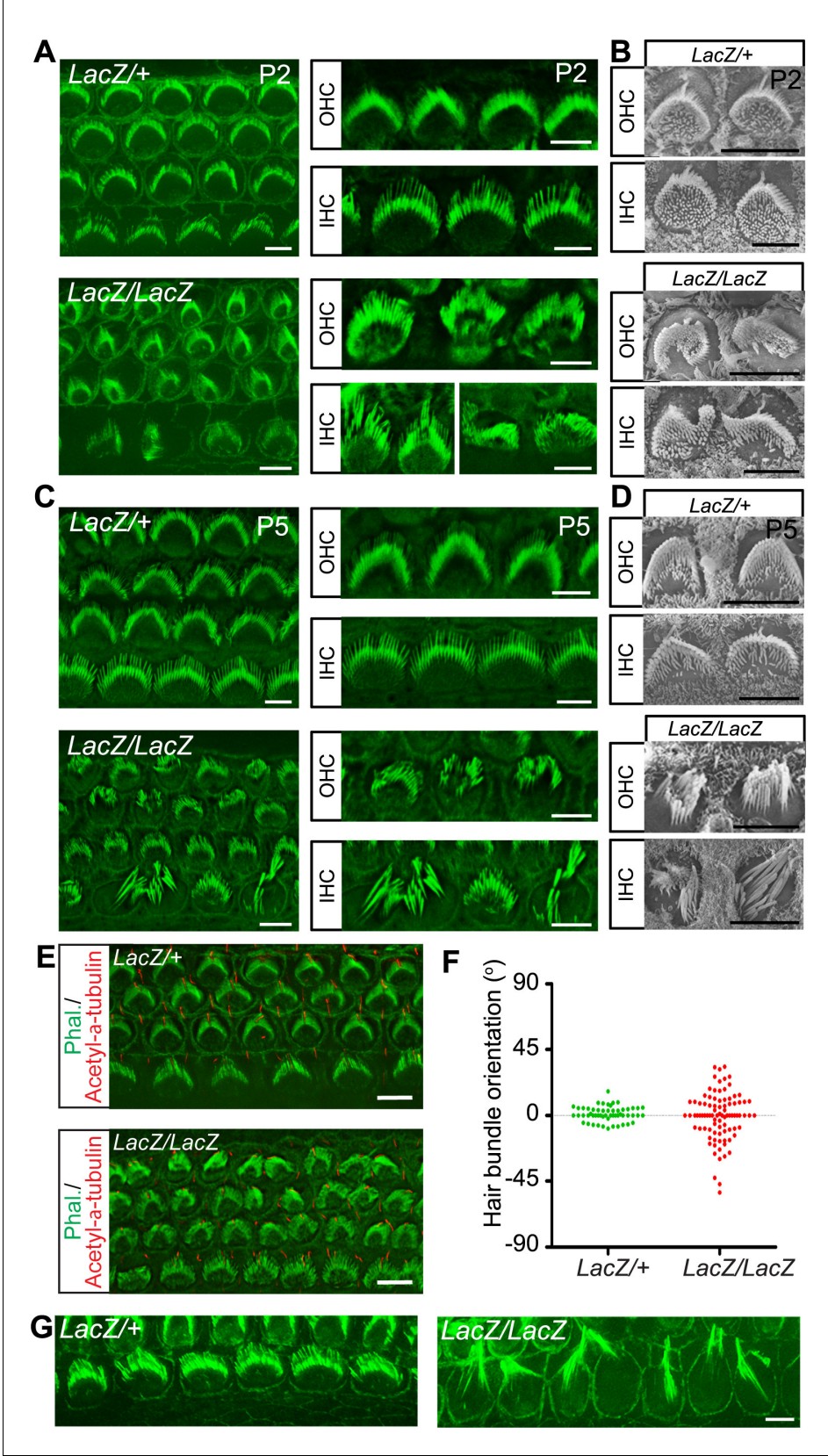

**Figure 3.** Hair cell morphology in *Fam65b*-deficient mice. (**A**, **C**) Analysis of the cochlea in P2 (**A**) and P5 (**C**) control and mutant animals by whole mount staining with phalloidin (green). Note the morphological changes in *Figure 3 continued on next page*

*Figure 3 continued*

hair bundles from mutant mice with defects in bundle polarity, cohesion and length of stereocilia. (B, D) Analysis of hair cells by SEM at P2 (**B**) and P5 (**D**). (**E**) Cochlear whole mounts at P2 were stained with phalloidin and with antibodies to acetylated–α–tubulin to reveal kinocilia. (**F**) Bundle orientation was determined by drawing a line through the axis of the bundle with 0° indicating the normal medio–lateral axis. Angular deviation from this axis was determined. Each dot represents one hair cell (n=55 for control; n=90 for mutants). (**G**) Cochlear explants from control and mutant animals were prepared at P2 and cultured for two days in vitro. Samples were stained by phalloidin. Note the hair bundles of most IHCs from mutants lacking Fam65b were disorganized and the stereocilia were extraordinarily long. Scale bars: 6 μm.

## Defects in mechanotransduction in *Fam65b*-deficient hair cells

Next we analyzed the extent to which mechanotransduction was affected in *Fam65b*-deficient hair cells. We stimulated hair bundles of OHCs at P5 with a stiff glass probe and recorded mechanotransduction currents in the whole-cell configuration. As previously reported, control OHCs from *Fam65-b$^{LacZ/+}$* mice had rapidly activating transducer currents, which subsequently adapted (*Gillespie and Müller, 2009*) (*Figure 6A*). Mechanotransduction currents could also be evoked in OHCs from *Fam65b$^{LacZ/LacZ}$* mice but peak currents were reduced (*Figure 6A*). The amplitude of saturated mechanotransduction currents at maximal deflection was at 417.16 ± 103.77 pA (mean ± SEM) for control OHCs and 232.60 ± 123.75 pA (mean ± SEM) for OHCs from mutant mice (*Figure 6B*). To normalize for variations in amplitude between control and mutant animals, we also plotted the open probability of the transduction channel ($P_0$) against displacement. The resulting curve in the mutant animals was slightly shifted to the right and broadened (*Figure 6C*). The resting potential was comparable in OHCs from control and mutant animals (data not shown), suggesting that the change in transduction and in the displacement- $P_0$ relationship was not a simple consequence of hair cell deterioration alone. However, given the structural defects in hair bundles of *Fam65b$^{LacZ/LacZ}$* mice, it seems likely that effects on transduction were caused at least in part by the morphological defects in hair bundles.

We have recently shown that injectoporation is useful to express the genetically encoded Ca$^{2+}$ sensor G-CaMP3 in hair cells and to then analyze mechanotransduction by changes in the fluorescence intensity of G-CaMP3 (*Xiong et al., 2012*; *Zhao et al., 2014*). To determine the extent to which defects in mechanotransduction in *Fam65b*-deficient hair cells could be rescued by re-expression of Fam65b, we injectoporated control and *Fam65b*-deficient hair cells at P2 with an expression vector for G-CaMP3 and a Fam65b construct carrying an HA-tag. After two days in culture, we mechanically stimulated hair bundles with a fluid jet with three consecutive pulses increasing in duration from 0.1 s, to 0.3 s and 0.5 s. Hair cells from control mice responded with a robust increase in fluorescence, while the response in *Fam65b*-deficient hair cells was significantly decreased (*Figure 6D,E*). Expression of HA-Fam65b in *Fam65b*-deficient hair cells restored transduction to similar levels as in controls (*Figure 6D,E*). We also visualized injectoporated hair cells with an antibody to HA to detect HA-Fam65b and visualized hair bundles by staining with phalloidin. Expression of HA-Fam65b did not affect the morphology of control hair cells but it rescued the morphological defects in hair cells from *Fam65b*-deficient mice (*Figure 6F*). These findings provide further evidence that Fam65b is required cell autonomously in hair cells to regulate their morphology and function, and show that functional defects can be repaired even in hair cells at early postnatal ages.

## Analysis of Fam65b and taperin distribution in hair cells

Super-resolution microscopy such as STORM has been successfully used to obtain higher spatial resolution of protein distribution in cellular compartments such as dendritic spines and immunological synapses than achievable with more conventional fluorescence microscopy techniques (*Xu et al., 2012*; *2013*). We attempted to apply this technique to mechanosensory hair cells. To analyze whether Fam65b and taperin are organized in a particular pattern at the base of stereocilia, we labeled cochlear whole mounts at P7 with antibodies to Fam65b and taperin. Following incubation with secondary antibodies coupled to Alexa Flour 647 we visualized protein distribution using STORM microscopy. The images revealed the localization of Fam65b and taperin at the base of stereocilia, but remarkably in distinct distribution patterns. At lower magnifications both Fam65b and

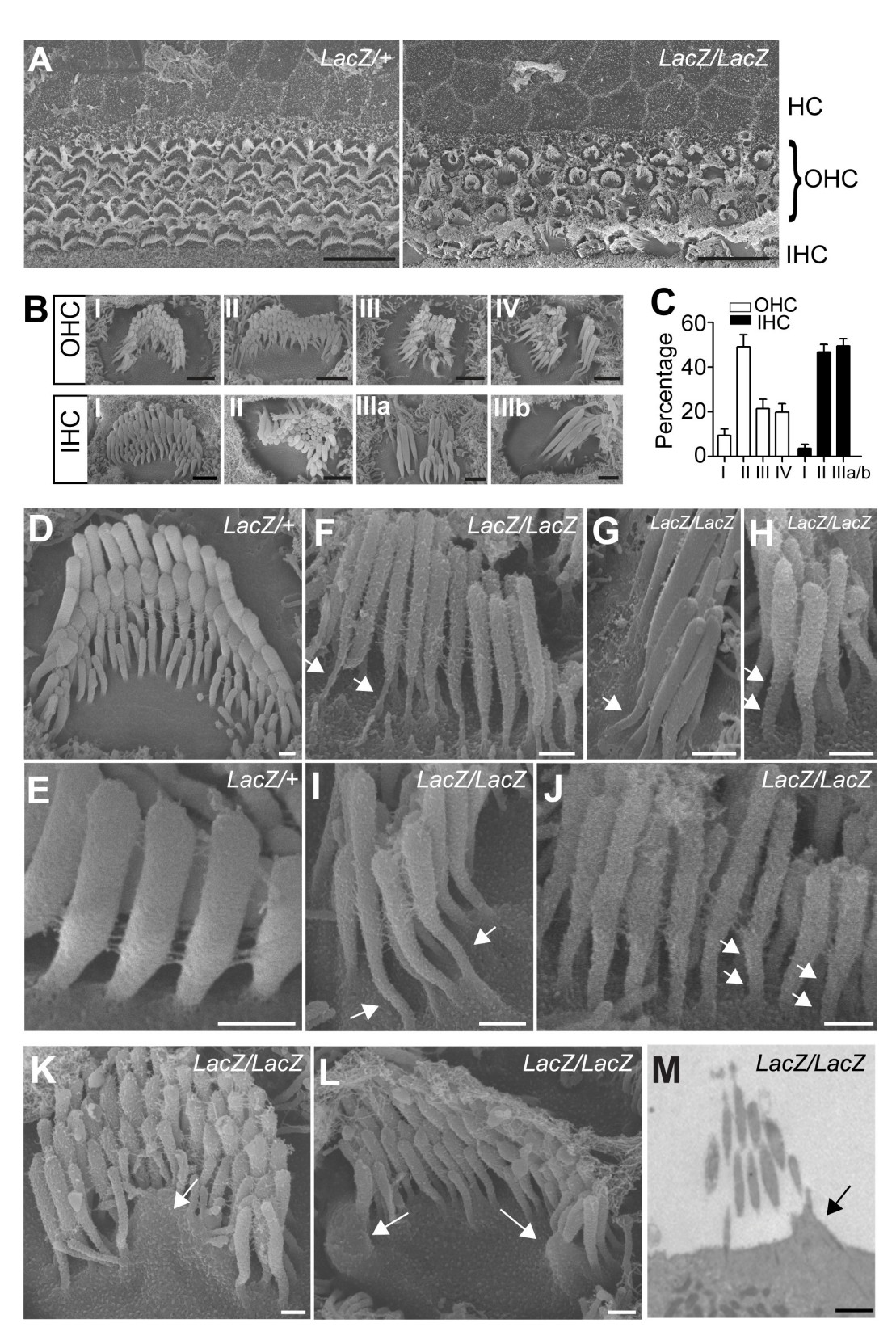

**Figure 4.** Analysis of hair bundle morphology by SEM. Whole mounts from the middle part of the cochlea from *Fam65b^{LacZ/+}*controls and *Fam65b^{LacZ/LacZ}*mutants at P5(**A**) where analyzed by SEM. (**A**) Low magnification view showing disorganization of hair bundles in the mutants. (**B**) Hair bundles were

*Figure 4 continued on next page*

*Figure 4 continued*

classified both for OHCs and IHCs according to morphological criteria. For OHCs: (I) rounded bundle–shape that deviated from the classical V–shape; (II) elongated but coherent bundles; (III) rounded bundles with signs of degeneration; (IV) fragmented bundles with some thin and elongated stereocilia. For IHCs: (I) coherent bundles with slight abnormalities; (II) coherent bundles with degenerative changes; (IIIa) fragmented bundles with thin and elongated stereocilia; (IIIb) fragmented bundles with few remaining thin and elongated stereocilia. (C) Quantification of the number of hair bundles in different morphological classes (mean ± SEM; n=503 for OHCs; n=191 for IHCs). (D–L) Higher magnification views of hair bundles. Arrows point to abnormally shaped basal domains of stereocilia (F–J) and to protrusions from the apical cell surface (K,L). (M) Transmission electron microscopy image. The arrow points to a protrusion from the apical cell surface. See also *Figure 4—figure supplement 1* for further data on the study of hair cell survival. Scale bars: (A) 10 µm; (B) 1 µm; (D–L) 0.3 µm(M) 2 µm.

The following figure supplement is available for figure 4:

**Figure supplement 1.** Lack of Fam65b interactions with taperin.

taperin outlined the general shape of the stereociliary bundle of OHCs and IHCs (*Figure 7A,D*), similar to deconvolution microscopy (*Figure 2*). At higher magnifications only accessible via STORM, Fam65b appeared as punctate dots that formed a circumferential ring outlining the edges of individual stereocilia (*Figure 7B,C*). In contrast, taperin appeared as dense puncta that extended into the core of stereocilia (*Figure 7E,F*). Fam65b ring-like structures were most clearly visible under the tallest row of stereocilia of IHCs. We therefore carried out measurements to determine the diameter of Fam65b ring-like structures and taperin circles in the tallest row of stereocilia of P7 inner hair cells using NIS-Elements AR analysis software.

The ring-like Fam65b structure had a diameter of 229 ± 7 nm (mean ± SEM; n=41), while the taperin structure had a diameter of 228 ± 9 nm (mean ± SEM; n=56). Thus, the two structures appear to overlap, although this is difficult to ascertain because the dimension of the primary and secondary antibody affects the measured dimension. As with deconvolution microscopy, both proteins could no longer be detected in the taper region in *Fam65b^{LacZ/LacZ}* mice (data not shown). These findings provide evidence that while both Fam65b and taperin are present near the taper of stereocilia, they show a distinct distribution pattern. Taperin is more widely expressed throughout the core of the stereocilium and Fam65b is present in a circumferential ring that might partially overlap with the distribution of taperin. Taperin is no longer concentrated near the taper region in *Fam65b*-deficient hair cells (*Figure 5*), which could be a direct consequence of effects of Fam65b on taperin or a secondary consequence caused by disruptions in the structure of the basal domain of stereocilia.

We next analyzed whether Fam65b might directly bind to taperin, or to CLIC5 and radixin, which are thought to form a protein complex with taperin (*Salles et al., 2014*). We also analyzed potential interactions with whirlin, a PDZ-domain protein that is localized near the base of stereocilia (*Grati et al., 2012*; *Michalski et al., 2007*), and with PTPRQ, which is also present in this domain of stereocilia (*Goodyear et al., 2003*). We cloned full-length cDNAs for taperin, CLIC5, radixin and whirlin, fused them to HA and co-expressed them with Fam65b-GFP in HEK293 cells. For PTPRQ, we expressed the cytoplasmic domain fused to HA. Immunoprecipitation experiments did not reveal any interactions between Fam65b-GFP with HA-taperin, HA-CLIC5, HA-radixin, HA-whirlin, and HA-PTPRQcyto (*Figure 7G*, and data not shown). To further confirm these findings, we also expressed HA-Fam65 and Myc-taperin in CL4 cells but did not observe an interaction between the two proteins (*Figure 7—figure supplement 1*). Similarly, we did not observe interaction between Fam65b and taperin in a two-hybrid assay where we used Fam65b as a bait and taperin as a prey (*Figure 7—figure supplement 1*). These results are consistent with the distinct localization pattern of Fam65b and taperin, and suggest a compartmentalized distribution of protein complexes within the basal compartment of stereocilia. They also suggest that the Fam65b regulates taperin localization via an alternative mechanism than direct binding. Unfortunately, we could not extend our analysis of protein distribution to CLIC5, radixin and whirlin because available antibodies were not of sufficient quality for STORM (data not shown).

## Fam65b forms oligomeric structures

The ring-like distribution of Fam65b near the taper of stereocilia suggested that Fam65b might form oligomeric structures. To determine the extent to which Fam65b oligomerizes we cloned full-length

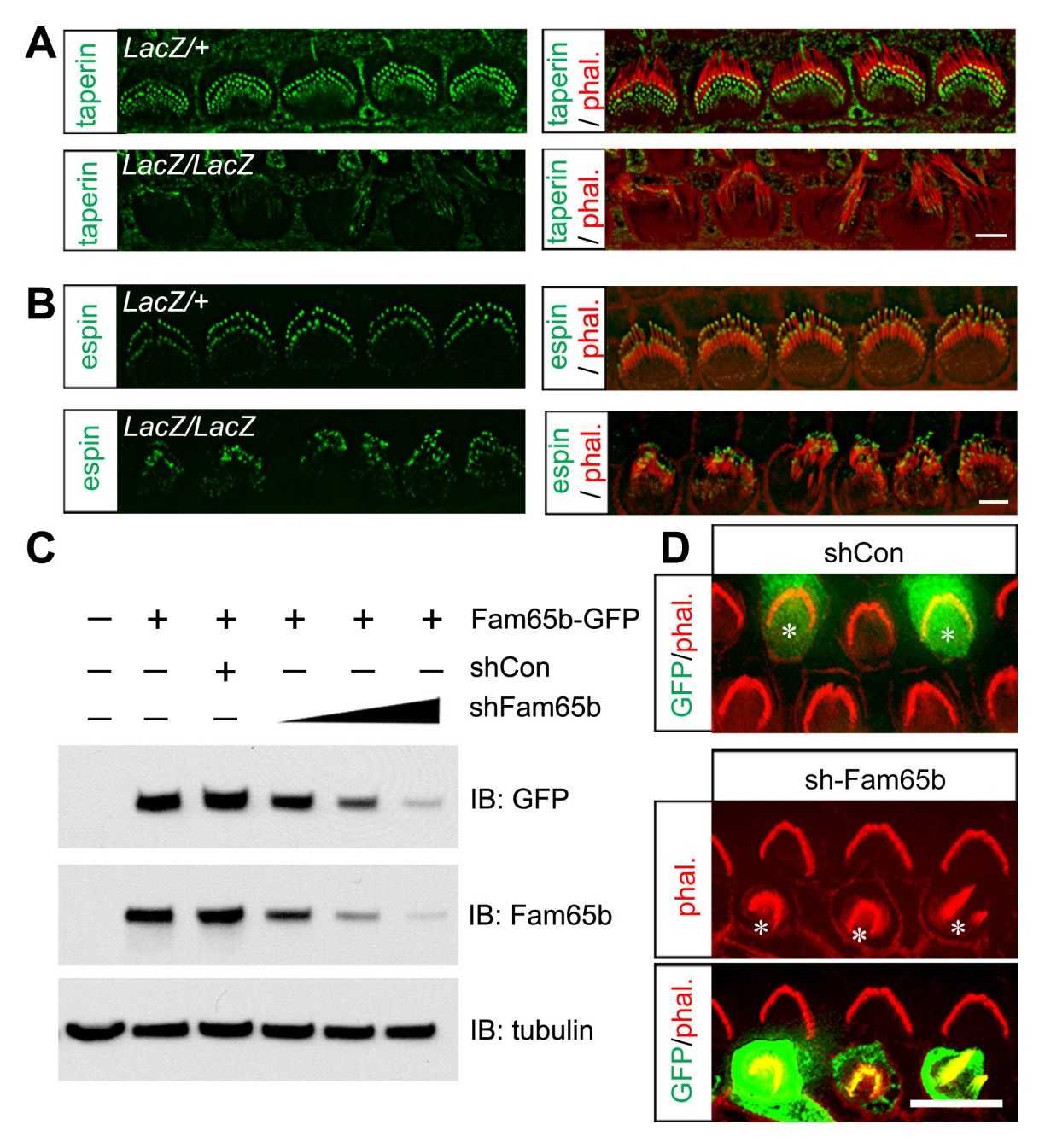

**Figure 5.** Taperin localization and shRNA perturbation of Fam65b. (**A, B**) Taperin (green, A) and espin (green, B) staining in control and mutant IHCs analyzed by fluorescent deconvolution microscopy. Stereocilia were visualized by staining with phalloidin–rhodamine (red). The taperin signal is reduced and diffused in mutant hair cells, while the espin signal is similar. (**C**) shRNA targeting Fam65b efficiently knocks down Fam65b–GFP expression in transfected HEK293 cells. Different ratios (2:1; 1:1 and 1:2) of Fam65b–GFP and shRNA plasmids were used. Scrambled shRNA was used as knock-down control and tubulin was used as a loading control for western blotting. (**D**) Cochlear explants were prepared at P2 and injectoporated to express GFP and control shRNA, or GFP and shRNA targeting Fam65b. Three days later, tissues were fixed and stereocilia were visualized by phalloidin staining. Note disorganized hair bundles in sh*Fam65b* positive hair cells (asterisks in middle panel) but not in hair cells expressing control shRNA (asterisks upper panel). Scale bars: (**A, B**) 3 μm; (**D**) 6 .

Fam65b as well as N- and C-terminal fragments (*Figure 8A*). First, we generated full-length Fam65b constructs containing N-terminal HA and C-terminal GFP tags and expressed them alone or in combination in HEK293 cells. We then carried out immunoprecipitations using an antibody to HA.

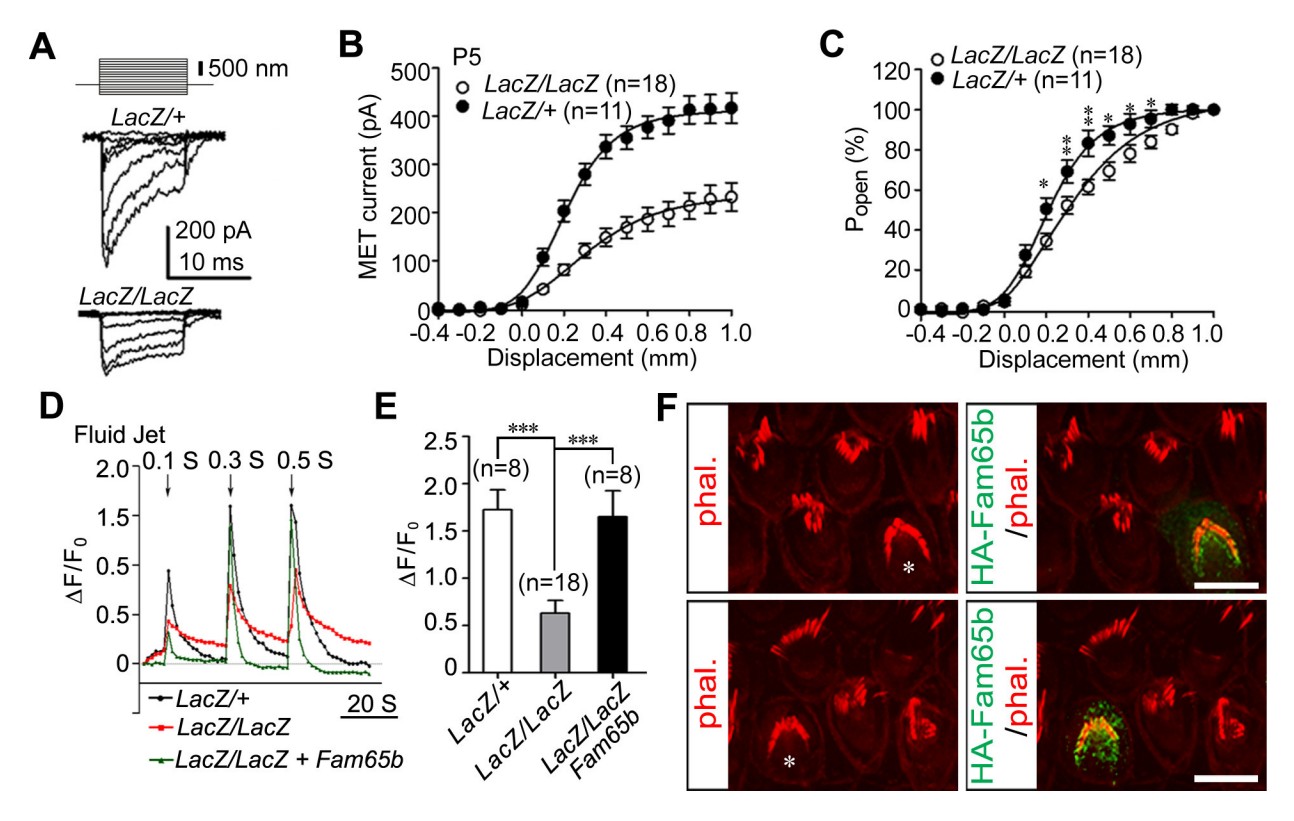

**Figure 6.** Analysis of mechanotransduction currents in *Fam65b*-deficient hair cells. (**A**) Examples of transduction currents in OHCs from control and mutant mice at P5 in response to a set of 10 msec hair bundle deflections ranging from –400 nm to 1000 nm (100 nm steps). (**B**) Current displacement plots obtained from similar data as shown in (**A**). (**C**) The $P_o$–displacement relationship plot obtained with peak currents following deflection reveals a significant rightward shift and broadening of the curve in mutant hair cells. (**D**) Representative example demonstrating fluid–jet induced $Ca^{2+}$ response in G-CaMP3-expressing OHCs from controls, *Fam65b* mutants, and *Fam65b* mutants following re-expression of Fam65b. OHCs were transfected at P2 and cultured for 2 days in vitro. Sequential fluid-jet pulse durations were 0.1 s, 0.3 s and 0.5 s. For quantitative analysis (panel E), the amplitude of the 2nd $Ca^{2+}$ response peak was measured. (**E**) Quantification of similar $Ca^{2+}$ responses as shown in (**E**). The number of analyzed hair cells is indicated in brackets. (**F**) *Fam65b*-deficient cochlea explants were prepared at P2 and injectoporated with HA-Fam65b. After in vitro culture for 2 days, samples were fixed and stereociliary morphology was visualized by phalloidin staining. Note the morphological defects could be rescued. Scale bar: 6 µm. All values are mean ± SE ***p<0.001, by Student's t-test.

Fam65b-GFP efficiently co-immunoprecipitated with HA-Fam65b (*Figure 8B*), demonstrating that Fam65b has homophilic binding affinity. To map protein domains important for interactions, we next generated truncated Fam65b constructs fused to a Myc-tag. Sequence comparison of Fam65b across species revealed five highly conserved domains near the N-terminus, two highly conserved domains near the C-terminus and a less conserved central domain (*Figure 8A*). We therefore generated Fam65b constructs containing the N-terminal or C-terminal conserved domains alone (Myc-N1, Myc-C2) or in conjunction with the less conserved central domain (Myc-N2, Myc-C1) (*Figure 8A*). Myc-N1 and Myc-C1 both interacted with full-length Fam65b-GFP (*Figure 8C*), indicating that there are at least two binding sites in Fam65b that mediate homophilic interactions. Myc-N1 efficiently interacted with N1-GFP and N2-GFP but not with C1-GFP or C2-GFP (*Figure 8D–G*). Likewise Myc-C1 efficiently interacted with C1-GFP and C2-GFP but not with N1-GFP or N2-GFP (*Figure 8D–G*). We thus conclude that Fam65b molecules can interact with each other via their N-termini and via their C-termini, which could lead to the formation of concatemers.

## Fam65b oligomerization is important for its function

To determine the extent to which oligomerization of Fam65b is important for its function, we wanted to disrupt interactions between Fam65b molecules in mechanosensory hair cells. We hypothesized

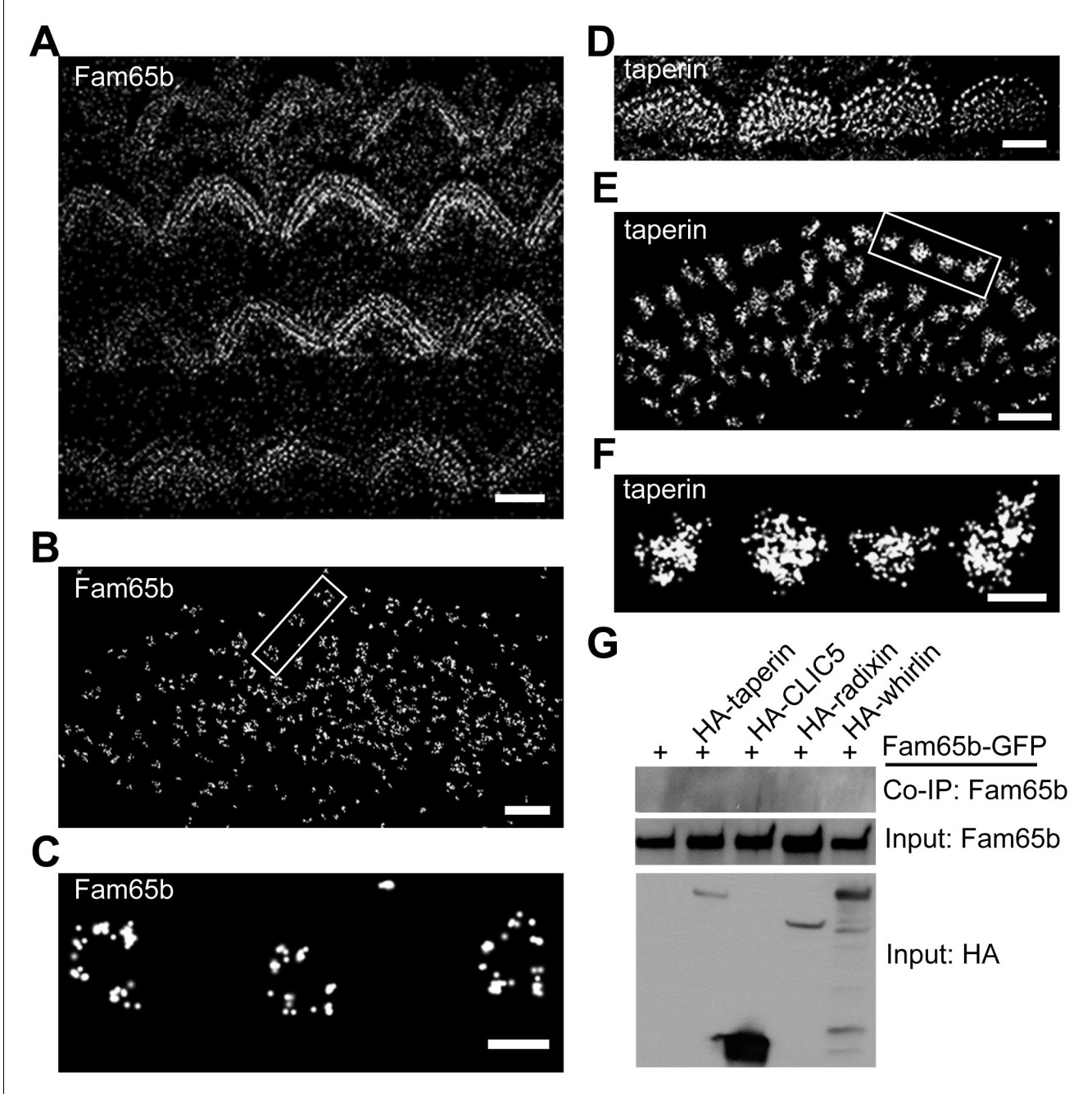

**Figure 7.** Distinct localization pattern of Fam65b and taperin resolved by STORM. (**A–F**) STORM images of Fam65b (**A–C**) and taperin (**D–F**) localization in hair cells. **A** and **D** show magnification/resolution similar to fluorescent deconvolution microscopy or traditional confocal microscopy. (**B, C, E,** and **F**) are higher magnifications only accessible by STORM, demonstrating the distinct localizations of Fam65b and taperin. (**C**) and (**F**) are close-up of boxed region in (**B**) and (**E**). (**G**) Lack of interactions between Fam65b and taperin, CLIC5, radixin and whirlin. HEK293 cells were transfected with the constructs indicated on top of each panel. Immunoprecipitations were carried out with HA antibody followed by western blotting to detect Fam65b-GFP. The lowest rows show input protein, the upper row shows co-immunoprecipitation (CoIP) results. See also *Figure 7—figure supplement 1* for further data on the analysis of interactions between Fam65b and taperin. Scale bars, (**A, D**) 3 µm; (**B, E**) 1 µm; (**C, F**) 200 nm.

The following figure supplement is available for figure 7:

**Figure supplement 1.** Minimal hair cell loss in *Fam65b* mutants at P16 and P28.

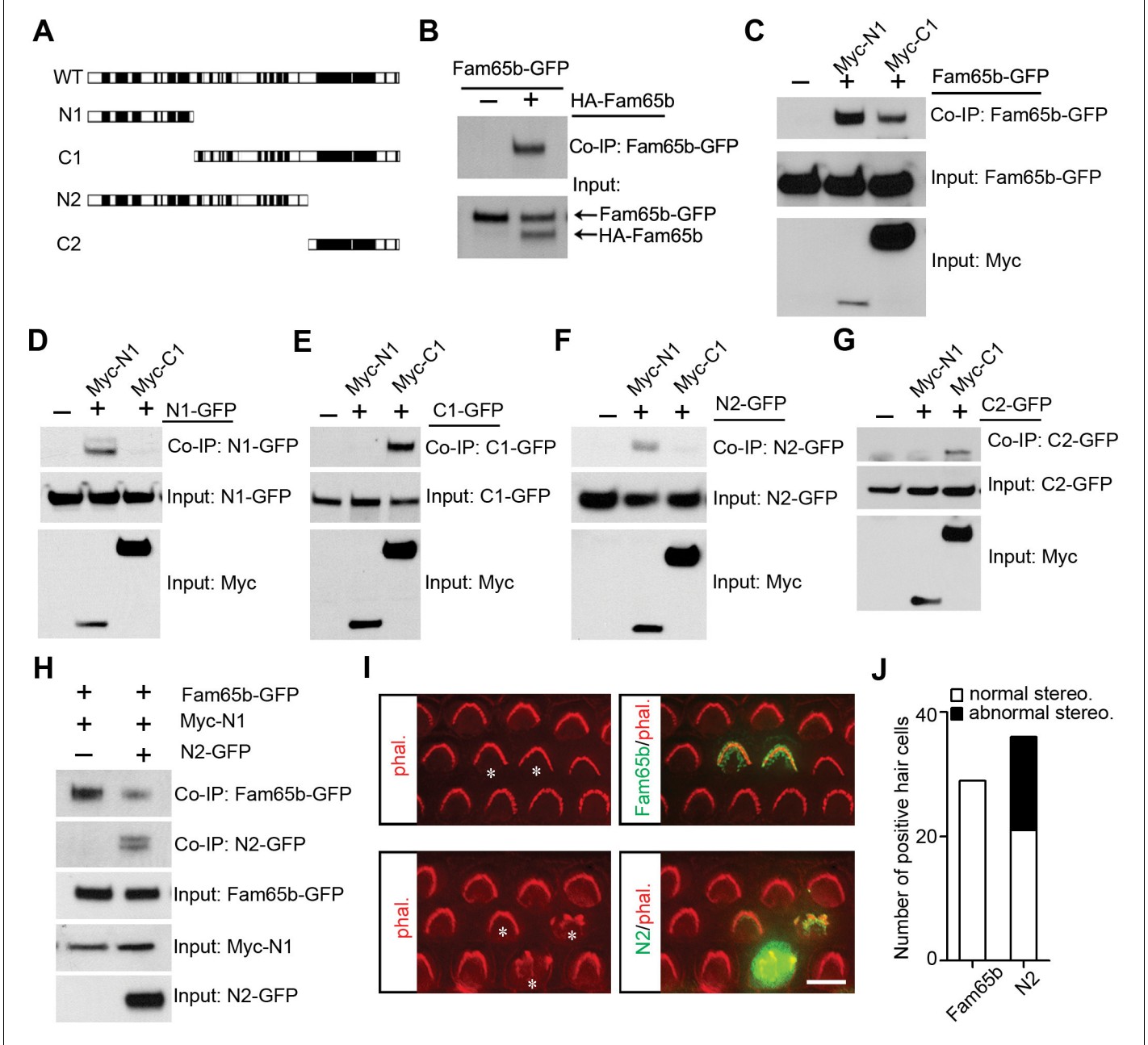

**Figure 8.** Fam65b forms oligomeric structures. (**A**) Diagram of the constructs used for biochemical experiments. Evolutionarily conserved domains are indicated in black. (**B–G**) HEK293 cells were transfected with the constructs indicated on top of each panel. Immunoprecipitations were carried out with HA (**B**) or Myc (**C–G**) antibodies, followed by western blotting to detect co–expressed proteins. The upper rows show CoIP results and the lower rows show input protein. (**H**) N2-GFP perturbs interactions between Fam65b-GFP and N2-GFP. CoIP blot results above and input protein blot results below. (**I**) OHCs were injectoporated at P2 to express Fam65b-GFP or N2-GFP, expression of which was evaluated 2 days later by immunohistochemistry (green). Stereociliary morphology of OHCs was visualized by phalloidin staining (red). Note disorganization of stereociliary hair bundles in N2-GFP transfected cells that are marked with an asterisk. Note that one hair cell in the lower panel expressed barely detectable levels of the transgene and showed no structural defects. (**J**) Number of transfected hair cells and quantification of morphology from (**I**). Black column represents hair cells with morphological defects. Scale bar: 6 μm.

that overexpression of a Fam65b fragment containing one Fam65b binding site might disrupt homo-philic interactions between Fam65b molecules thus acting as a dominant negative construct. To test this model, we first determined the extent to which interactions between full-length Fam65b-GFP

with Myc-N1 was disrupted by co-expression of N2-GFP in heterologous HEK293 cells. Co-immuno-precipitation experiments demonstrated that N2-GFP significantly reduced interactions between Fam65b-GFP and Myc-N1 (*Figure 8H*). Next we expressed by injectoporation full-length Fam65b-GFP or the dominant negative construct N2-GFP in hair cells at P2 and cultured the cells for 3 days. Cells expressing the heterologous constructs were identified by GFP fluorescence and hair bundles were visualized by staining with phalloidin-rhodamine. Significantly, while Fam65b-GFP did not affect the morphology of hair bundles (*Figure 8I,J*), bundle morphology was severely disrupted by overexpression of N2-GFP (*Figure 8I,J*). The data suggest that homophilic interactions between Fam65b molecules are required for Fam65b function in hair cells, although we cannot exclude that N2-GFP might have affected interactions with other proteins as well.

## Fam65b binds to RhoC

Our co-immunoprecipitation data demonstrated that Fam65b does not interact with taperin or with other proteins that have been localized near the base of stereocilia such as CLIC5, radixin, and whirlin (*Figure 7C*). We therefore set out to identify binding partners for Fam65b that might regulate its function and oligomerization using an unbiased yeast-two-hybrid screen. We purified RNA from the organ of Corti and generated yeast-two-hybrid libraries suitable for the identification of soluble and transmembrane proteins. We then carried out yeast-two-hybrid screens with full-length Fam65b. We screened more than 2 million transformants and identified 50 positive clones. Sequencing of the clones revealed that we had recovered RhoC 28 times and RhoA 1 time (data not shown). In fact, RhoA has been previously shown to be a binding partner for Fam65b (*Gao et al., 2015*; *Rougerie et al., 2013*), but interactions with RhoC have not been evaluated so far. To confirm our yeast-two-hybrid data, we generated HA-tagged versions of RhoA, RhoC and the related Rac1 and Cdc42 molcules and tested for interactions with Fam65b following their expression in HEK293 cells. Fam65b bound highly efficiently to RhoC, weakly to RhoA, and not at all to Rac1 and Cdc42 (*Figure 9A*). Taken together, these data provide evidence that Fam65b binds strongly to RhoC but much weaker to RhoA.

To determine the extent to which RhoC and Fam65b might functionally interact in vivo, we wanted to determine the subcellular distribution of RhoC in mechanosensory hair cells. Unfortunately, available antibodies for RhoC were not of sufficient quality for immunolocalization studies. We therefore expressed by injectoporation HA-tagged RhoC and Fam65b-GFP in mechanosensory hair cells at P2 and evaluated their localization two days later. Stereocilia were visualized by staining with Alexa Fluor 647 phalloidin. Remarkably, RhoC was localized near the base of stereocilia and its distribution largely overlapped with the distribution of Fam65b (*Figure 9B*). We thus conclude that RhoC binds to Fam65b and shows a similar distribution in mechanosensory hair cells, at least at the level of resolution provided by deconvolution microscopy. Once suitable antibodies become available, it will be important to further refine the distribution of RhoC in hair cells using STORM microscopy.

## RhoC regulates Fam65b oligomerization

Previous studies in immune cells have suggested that Fam65b might regulate RhoA activity (*Gao et al., 2015*; *Rougerie et al., 2013*). Fam65b might therefore act in hair cells upstream of RhoC to regulate its activity. Alternatively, RhoC might regulate Fam65b function. Rho-GTPases cycle between the active GTP bound state and inactive GDP bound state (*Dvorsky and Ahmadian, 2004*). Activated RhoC binds effector proteins such as Rhotekin. Following expression of Fam65b and RhoC in heterologous cells, we did not observe obvious inhibitory effect of Fam65b on interactions of RhoC with Rhotekin, suggesting that Fam65b does not regulate RhoC activity (data not shown). We therefore tested next whether RhoC might instead affect Fam65b function and determined the extent to which RhoC regulates Fam65b oligomerization. We expressed RhoC together with full-length Fam65b-GFP and Myc-Fam65b in HEK293 cells. Fam65b complexes were recovered by immunoprecipitation using a Myc antibody. Significantly, homophilic interactions between Fam65b molecules were strongly enhanced by RhoC (*Figure 9C,D*).

Next, we expressed full-length Fam65b-GFP with N-terminal (Myc-N1) and C-terminal (Myc-C1) fragments of Fam65b either alone or together with RhoC. Interactions between Fam65b-GFP and Myc-N1 but not Myc-C1 were strongly enhanced by RhoC (*Figure 9E,F*), suggesting that RhoC

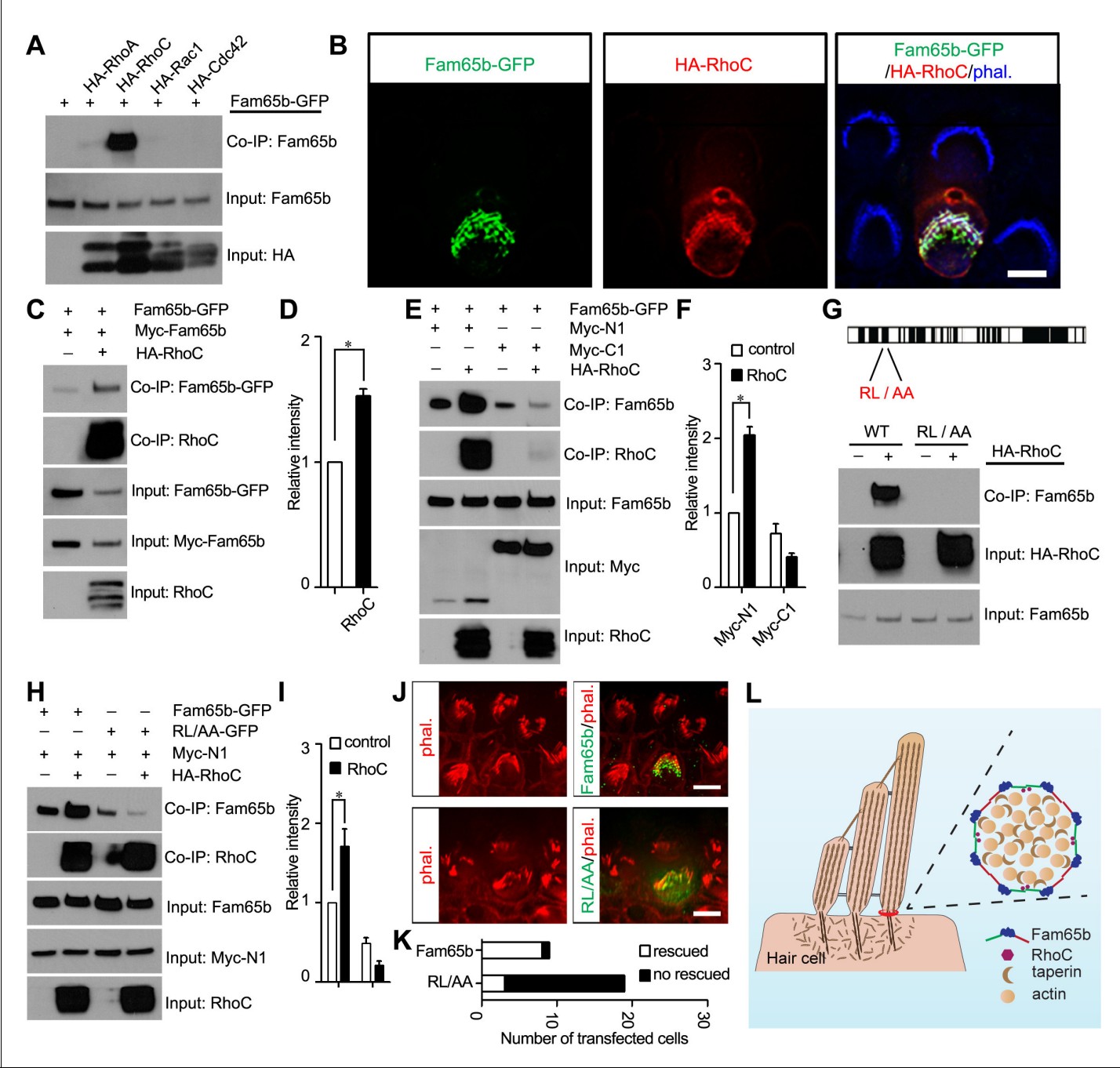

**Figure 9.** Effects of RhoC on Fam65b oligomerization. (**A**) HEK293 cells were transfected with the constructs indicated on top of each panel. Immunoprecipitations were carried out with HA antibody, followed by western blotting to detect co–expressed proteins. The upper row shows CoIP result and the lower rows show input protein. Note strong binding activity was detected between Fam65b and RhoC, weak binding was detected with RhoA, while no binding was detected with Rac1 and Cdc42. (**B**) OHCs were injectoporated at P2 to express Fam65b–GFP and HA–RhoC. Expression of Fam65b-GFP (green) and HA-RhoC (red) was evaluated 2 days later by immunohistochemistry. Stereocilia were visualized by phalloidin staining (blue). Note colocalization of Fam65b-GFP and HA-RhoC at the base of stereocilia. Scale bar: 4 μm. (**C, E, G, H**) HEK293 cells were transfected with the constructs indicated on top of each panel. Immunoprecipitations were carried out with Myc antibody (**C, E, H**) or HA antibody (**G**), followed by western blotting to detect co-expressed proteins. The upper rows show CoIP results and the lower rows show input protein. Note co-expression of HA-RhoC enhances/stabilizes the oligomerization of Fam65b at N termini (**C**) but not C termini (**E**). RhoC did not promote oligomerization of a Fam65b mutant (RL changed to AA; diagram in **G**) devoid of RhoC binding activity (**G, H**). (**D, F, I**) Quantification of CoIP results by scanning of similar gels as shown in (**C**) (**E**) and (**H**). The values are derived by quantifying more than 3 independent experiments. (**J**) *Fam65b*-deficient OHCs were injectoporated at P2 to express Fam65b-GFP or mutant Fam65b protein without RhoC binding activity (Fam65b$^{RL/AA}$-GFP). Expression of Fam65b was evaluated 2 days later by

*Figure 9 continued on next page*

*Figure 9 continued*

staining for GFP (green). Stereociliary morphology of OHCs was visualized by phalloidin staining (red). Note morphological defects of stereocilia were rescued by Fam65b-GFP but not by Fam65b$^{RL/AA}$-GFP. Scale bar: 6 μm. (K) Quantification of transfected hair cells from (J). White column represents number of rescued OHCs. (L) Model of Fam65b localization pattern in hair cells. In our model, Fam65b forms a circumferential ring near the basal taper domain of stereocilia, while taperin forms a dense core structure. Fam65b forms oligomers via head-to-head and tail-to-tail interactions and RhoC binds to Fam65b and promotes oligomerization.

facilitates interactions between the N-terminal of Fam65b. These data suggest that Fam65b acts downstream of RhoC, where RhoC regulates Fam65b oligomerization by promoting interactions between the N-termini of Fam65b molecules.

## RhoC binding to Fam65b is critical for Fam65b function in hair cells

To determine the extent to which binding of Fam65b to RhoC is critical for its function in hair cells, we generated a mutant Fam65b construct carrying two point mutations (Fam65b$^{RL/AA}$) that are thought to be critical for Fam65b function (*Figure 9G*) (*Gao et al., 2015*). Notably, the RL/AA mutation significantly abolished binding of Fam65b to RhoC (*Figure 9G*). We next analyzed the extent to which the RL/AA mutation affected the ability of RhoC to regulate Fam65b homophilic interactions. We expressed RhoC together with full-length Fam65b$^{RL/AA}$-GFP and the N-terminal Fam65b fragment Myc-N1 in HEK293 cells and carried out co-immunoprecipitation experiments using a Myc antibody. As predicted, while RhoC strongly enhanced homophilic interactions between wild-type Fam65b-GFP and Myc-N1, it no longer enhanced interactions between Fam65b$^{RL/AA}$-GFP and Myc-N1 (*Figure 9H,I*). In fact, the mutation appeared to even reduce the basal homophilic binding activity of Fam65b that is observed without expression of RhoC (*Figure 9H,I*).

To evaluate the importance of interactions of RhoC with Fam65b for hair cell function, we compared the ability of wild-type Fam65b with Fam65b$^{RL/AA}$ to rescue the morphological hair bundle defect of *Fam65b*-deficient hair cells. We expressed Fam65b-GFP or Fam65b$^{RL/AA}$-GFP by injectoporation in OHCs at P2 and analyzed hair bundle morphology 2 days later. The quantification of the data was facilitated by the fact that morphological defects in hair bundles from mutant mice are exaggerated in vitro, presumably because the bundles are fragile and more easily disrupted during the culture period compared to wild-type. Thus most hair bundles in cultured OHCs are either round or fragmented, which is in stark contrast to the V-shape of the bundles of OHCs in wild-type. Following injectoporation, Fam65b-GFP was efficiently targeted to the basal region of hair cell stereocilia and rescued the morphological hair bundle defect of OHCs from *Fam65b*-deficient mice, which formed the typical V-shape (*Figure 9J,K*). In contrast, Fam65b$^{RL/AA}$ could not rescue the morphological hair bundle defect in hair cells from *Fam65b*-deficient mice (*Figure 9J,K*). We thus conclude that binding of Fam65b to RhoC is critical for its function in mechanosensory hair cells.

## Discussion

The basal domain of stereocilia has been recognized as a specialized compartment with a distinct lipid and protein composition and a critical function in mechanotransduction because stereocilia form at their base a taper that allows them to pivot in response to mechanical stimulation (*Fettiplace and Kim, 2014*; *Müller and Barr-Gillespie, 2015*; *Zhao and Müller, 2015*). Using STORM microscopy we provide here insights into the spatial arrangement of cytosolic proteins within the basal domain of stereocilia at a resolution that had not been possible with other more common fluorescence microscopy techniques. These imaging experiments combined with biochemical and functional studies also provide insights into the function of Fam65b, a protein that has been previously linked to inherited forms of deafness but whose molecular and cellular function has remained unknown. Our findings provide evidence that Fam65b oligomers form a circumferential ring-like structure in the basal domain of each stereocilium and that this structure is critical for the maintenance of the morphology of the hair bundle. In the absence of the Fam65b the taper region appears abnormally shaped and hair bundles start to form but show degenerative changes before they reach maturity. Oligomerization of Fam65b depends on binding of Fam65b to RhoC, which co-localizes with Fam65b in the basal domain of stereocilia. In contrast, taperin binds to distinct

proteins and is distributed within the basal domain through the core of each stereocilium. Taperin distribution is disrupted in the *Fam65b*-deficient hair cells suggesting a potential functional interaction between taperin and Fam65b, but our biochemical data suggest that Fam65b and taperin do not interact directly. We therefore conclude that Fam65b has a critical function in the maintenance of the basal domain of stereocilia, and that defects in this process likely are the underlying cause of deafness in humans and mice with mutations in the *Fam65b* gene.

Previous studies have demonstrated that a mutation in *Fam65b* can lead to non-syndromic, pre-lingual, profound hearing loss (*Diaz-Horta et al., 2014*). We show here that *Fam65b*-deficient mice similarly are deaf already at 4 weeks of age, the first time point analyzed. Studies with morpholinos directed against Fam65b in zebrafish suggest that Fam65b is required for normal hair cell survival. However, we did not observed defects in hair cell survival at 2 and 4 weeks after birth, suggesting that the death of hair cells is not the primary cause of the auditory phenotype in mice lacking Fam65b. The difference in cell survival observed in the previous and current study may be attributed to species-specific functions for Fam65b. In *Fam65b*-deficient mice, we observe defects in hair bundle morphology already in the first few days after birth suggesting that degenerative changes are manifest prior to the onset of hearing in mice. The structural defects are associated with defects in mechanotransduction, indicating functional defects in hair cells that likely cause the auditory phenotype of the mutant mice. However, it is unclear whether the defects are caused by direct effects of Fam65b that affect the transduction process, for example by affecting the stiffness of stereocilia, or if the defects are a secondary consequence of the morphological hair-bundle defects that might affect transduction secondarily, for example by affecting the integrity of tip links. Further studies with more subtle mutations in Fam65b that do not dramatically affect hair bundle morphology might be informative.

While our findings suggest that defects in hair cells are critical for the auditory phenotype caused by *Fam65b* mutations, we also observed Fam65b expression in the microvilli of supporting cells including Hensen's cells and it cannot be excluded that defects in their development and function might contribute to the auditory phenotype. Regardless of the involvement of supporting cells, defects in hair bundles of hair cells are likely cell-autonomous because we can induce them by knock-down of Fam65b expression in hair cells (*Figure 4*) and rescue them by re-expression of Fam65b in *Fam65b*-deficient hair cells (*Figure 5*). The acute rescue of hair cell morphology by re-expression of Fam65b at postnatal ages also suggests that defects in hair bundle morphology are likely not a secondary consequence of perturbations in an overall developmental program but are more specifically linked to a role of Fam65b in stereocilia.

Previous studies have suggested that Fam65b might be a membrane-associated protein of stereocilia. Immunolocalization studies demonstrated that Fam65b was expressed along the entire length of stereocilia but additional staining was observed as punctate aggregates in the cell body (*Diaz-Horta et al., 2014*). In some of the hair cells analyzed by *Diaz-Horta et al. (2014)*, labeling appeared more prominent near the base of stereocilia, which is consistent with the expression pattern that we describe here. One possible explanation for the discrepancy in the findings is that the antibodies used by Diaz-Horta were raised against the N-terminus of Fam65b, which shows 70% similarity with Fam65a and Fam65c. We used an antibody to the C-terminus of Fam65b, which has very low homology with Fam65a/c. Perhaps hair cells express Fam65a,b,c and *Diaz-Horta et al. (2014)* observed with their antibodies the expression of all three Fam65 proteins. Clearly, our antibody appeared to be specific for Fam65b because we observed no staining in *Fam65b*-deficient hair cells. It will be interesting to define in the future the extent to which Fam65a and Fam65c are expressed in hair cells.

Given the diameter of stereocilia, the resolution of the images presented previously (*Diaz-Horta et al., 2014*) was clearly also not sufficient to distinguish between a localization of Fam65b at the membrane of stereocilia, localized internally within their core, or both. Membrane-targeting was mainly inferred previously because sequence comparison indicated that Fam65b might have a PX-BAR domain that can associate with membranes (*Diaz-Horta et al., 2014*; *Farooq and Tekin, 2014*). However, this conclusion was subsequently contested (*Teasdale and Collins, 2014*). Using STORM microscopy, we have now refined the localization of Fam65b within stereocilia and demonstrate that it is concentrated at their basal domain and not localized uniformly along the entire length of stereocilia. In addition, we show that Fam65b is localized in a circumferential ring near the stereociliary base and does not penetrate the core of stereocilia. The distribution of Fam65b in hair cells would

be consistent with localization at the membrane but it is currently not clear whether Fam65b directly binds to the membrane or is targeted to its location within stereocilia by interaction with other proteins. One intriguing possibility is that Fam65b might contain a modified PX-BAR domain that facilitates interactions with the specific lipids that have been shown to be concentrated at the stereociliary base (*Zhao et al., 2012*).

Our findings also provide insights into the mechanisms by which Fam65b acts in hair cells. Previous studies have shown that Fam65b can bind to RhoA and inhibit its function (*Gao et al., 2015*; *Rougerie et al., 2013*). We show here that Fam65b binds much more strongly to RhoC than to RhoA but we did not observe effects of Fam65b on RhoC activity. Instead, we show that RhoC co-localized with Fam65b in stereocilia and regulates Fam65b oligomerization. Oligomerization is driven by interactions between the N-termini of Fam65b, which are promoted by RhoC, and by interactions between the C-termini suggesting that Fam65b monomers can assemble into extended filaments. In fact, STORM imaging of Fam65b following its expression in heterologous cells (data not shown) suggests that oligomerization leads to linear filaments that can also bend into ring-like structures similar to those observed in hair cells. One interesting possibility would be that Fam65b oligomers form spring-like structures near the membrane at the base of stereocilia that affect the mechanical properties of the bundle. We are currently carrying out structural studies to address this possibility.

Intriguingly, we did not observe in our assays interactions of Fam65b with other cytoplasmic proteins that have been localized to the basal domain of stereocilia such as taperin, CLIC5, radixin and whirlin. While we cannot exclude that these proteins interact in some context, for example when they carry hair-cell specific posttranslational modifications, the findings nevertheless suggest the existence of functionally distinct protein complexes near the basal taper of stereocilia. Our STORM imaging data further provide evidence that taperin has a distribution in hair cells different from Fam65b. While both proteins are localized near the base of stereocilia, only taperin is localized within the core of stereocilia. Fam65b instead forms a circumferential ring. The diameter of the domain occupied by taperin and of the Fam65b ring are similar, raising the possibility that the two compartments show partially overlap. Unfortunately, antibodies to CLIC5 and radixin were not of sufficient quality to use them in STORM microscopy, but one might predict that CLIC5 and radixin, which are thought to form a protein complex with taperin (*Salles et al., 2014*), have a similar distribution at the base of stereocilia as taperin. Taperin localization is perturbed in the absence of Fam65b, suggesting that the distribution of protein complexes containing Fam65b and taperin is interdependent. Perhaps the two protein complexes are connected by a linker protein that still needs to be identified. Alternatively, the proteins might be components of distinct molecular complexes with separate function. Regardless, our data strongly suggest that Fam65b expression and oligomerization is critical for maintaining the structural organization of the basal domain of stereocilia and that mutations in Fam65b cause deafness at least in part by affecting stereocilia morphogenesis and function. Re-expression of Fam65b in *Fam65b*-deficient hair cells at postnatal ages rescued their morphological defects, suggesting that it might be possible to repair hair cell damage in human patients with mutations in the *FAM65B* gene.

In summary, our data highlight the power of STORM microscopy, combined with genetic and biochemical approaches, to provide insights into the mechanisms by which proteins regulate hair cell function. Notably, the stereocilia of hair cells contain hundreds of proteins many of which are affected in disease (*Krey et al., 2015*; *Petit and Richardson, 2009*; *Shin et al., 2013*; *Wilmarth et al., 2015*). However, stereocilia are small in size and it is challenging to define the function of individual proteins without precise spatial information. Our data show that STORM microscopy can be applied successfully to hair cells and will thus be a useful tool to investigate the spatial organization of proteins in stereocilia in normal and pathological conditions.

## Materials and methods

### Mouse strains, ABR and DPOAE measurement

ES cells carrying a *Fam65b* allele in which exons 2–14 of *Fam65b* were substituted by a LacZ expression cassette were obtained from Regeneron Pharmaceuticals, Inc. and used to generate *Fam65b*$^{LacZ/+}$ mice by the KOMP Repository and the Mouse Biology Program at the University of

California, Davis. *Fam65b*$^{LacZ/+}$ mice were maintained on a C57BL/6 background. ABR and DPOAE measurements were carried out as described (*Schwander et al., 2007*). All of the animal experiments were approved by the ethics committee of the institutional animal care and use committees of The Scripps Research Institute (Approval number 080088).

## LacZ staining
LacZ staining was performed as described (*Zhao et al., 2014*). In brief, tissue was fixed for 1 hr in PFA and incubated for 2 days at 4°C in 20% sucrose/PBS. Cryosections or acutely dissected tissues were fixed for 15 min at room temperature in 1% PFA, 0.2% glutaraldehyde, 0.02% NP40 and 0.01% sodium deoxycholate, and washed 3 times in PBS containing 0.02% NP40 and 0.01% sodium deoxycholate. Sections were stained overnight in the 1 mg/ml X-Gal staining solution (25 mM potassium ferricyanide, 25 mM potassium ferrocyanide, 2 mM $MgCl_2$, 1 mg/ml X-Gal diluted in PBS) at 37°C. Sections were washed 3 times for 20 min in PBS and post-fixed overnight at 4°C in 4% PFA. Sections were washed in distilled water, dehydrated and mounted.

## Whole mount staining and super-resolution imaging
Cochlear whole mount staining was carried out as described (*Zhao et al., 2014*). Whole mount images were captured by fluorescent deconvolution microscopy (Deltavision). Measurements of hair-bundle polarity were carried out as described (*Webb et al., 2011*). Whole mount staining for super-resolution imaging was performed according to the literature (*Xu et al., 2013*; . In brief, samples were incubated with primary antibodies followed by incubation with Alexa 647-conjugated secondary antibodies. Samples were mounted in freshly prepared STORM buffer (50 mM Tris, pH8.0, 10 mM NaCl, 10% glucose, 0.1 M mercaptoethanolamine [cysteamine; Sigma], 56 U/ml glucose oxidase [Sigma] and 340 U/ml catalase [Sigma]) and imaged on a Nikon N-STORM on a Ti-E inverted microscope. Samples were imaged using a 100X/1.49 NA Apo TIRF objective. Images were collected on an Andor IXON3 Ultra DU897 electron-multiplying CCD camera using the multicolor continuous mode setting in the Nikon Elements software. Power on the 647-nm lasers was adjusted to enable collection of 20–300 molcules per 256 X 256 camera pixel frame at appropriate threshold setting. More than 10,000 frames were collected and reconstructed with Nikon STORM software.

Primary antibodies were as follows: α-Fam65b (rabbit, Sigma); α-HA (mouse, Cell signaling); α-Myc (rabbit, Cell signaling); α-taperin (rabbit, Sigma); α-espin (Mo, BD Transduction Laboratories); anti-acetylated-α-tubulin (mouse, Sigma, 6-11B-1). Additional reagents were: Alexa Fluor 488-phalloidin, Alexa Fluor 647-phalloidin (Invitrogen, Carlsbad, CA), Alexa Fluor 488 and Alexa Fluor 647 conjugated goat anti-rabbit secondary antibodies.

## Scanning electron microscopy
The experiment was performed as described (*Zhao et al., 2014*). In brief, inner ears were dissected in fixative (2.5% glutaraldehyde; 4% formaldehyde; 0.05 mM Hepes Buffer pH 7.2; 10 mM $CaCl_2$; 5 mM $MgCl_2$; 0.9% NaCl). Sample was fixed for 2 hr at RT, and dissected in washing buffer (0.05 mM Hepes Buffer pH 7.2; 10 mM $CaCl_2$; 5 mM $MgCl_2$; 0.9% NaCl). The stria vascularis, Reissner's membrane and tectorial membrane were removed. Samples were dehydrated and processed to critical drying point in an Autosamdri-815A (Tousimis). Cochlea were mounted with carbon tape and coated with iridium (sputter coater EMS150TS; Electron Microscopy Sciences). Samples were imaged with a Hitachi S-4800-ll Field Emission Scanning Electron.

## DNA constructs, transfections, immunoprecipitations and western blots
DNA constructs are described in Supplementary Information. Expression of the constructs, immunoprecipitations, and western blots were carried out as described (*Senften et al., 2006*). Immunoprecipitation experiments were carried out at least 3 times to verify the reproducibility of the data. The following antibodies were used for the experiments: α- Fam65b (rabbit, Sigma); α-HA (mouse, Cell signaling); α-Myc (rabbit, Cell signaling); α-GFP (*Xiong et al., 2012*); α-GFP (mouse, Santa Cruz).

## Injectoporation

The experiment was performed as described (*Xiong et al., 2012*; *Zhao et al., 2014*). In brief, the organ of Corti was isolated and placed in DMEM/F12 medium with 1.5 µg/ml ampicillin. For electroporation, glass electrodes (2 µm diameter) were used to deliver plasmid (500 ng/µl in 1x HBSS) to the sensory epithelium. A series of 3 pulses was applied at 1 s intervals with a magnitude of 60V and duration of 15 msec (ECM 830 square wave electroporator; BTX). For $Ca^{2+}$ imaging, we used G-CaMP3 (Addgene 22692). Hair cells were imaged on an upright Olympus BX51WI microscope mounted with a 60x water-immersion objective and Qimaging ROLERA-QX camera, controlled by Micro-Manager 1.3 software (*Edelstein et al., 2010*). Hair bundles were stimulated with a fluid jet applied through a glass electrode (2 µm tip-diameter) filled with bath solution. Stimuli were applied using Patchmaster 2.35 software (HEKA) and 20 psi air pressure. Images were collected with a 2 s sampling rate. A series of fluid-jet stimulations (0.1, 0.3, 0.5 s) was applied (60 s intervals). Responses induced by 0.3 s fluid-jet stimulation were used for quantitative analysis.

## Electrophysiology

The experiment was performed as described (*Zhao et al., 2014*). In brief, during recording, a Peri-Star peristaltic pump (WPI) was used to perfuse artificial perilymph (in mM): 144 NaCl, 0.7 $NaH_2PO_4$, 5.8 KCl, 1.3 $CaCl_2$, 0.9 $MgCl_2$, 5.6 glucose, and 10 H-HEPES, pH 7.4. In some recordings, $Ca^{2+}$ concentration was reduced to 0.02 mM. To record reverse polarity currents in OHCs from wildtype animals, 5 mM BAPTA was added to the bath solution. Borosilicate glass with filament (Sutter) was pulled with a P-97 pipette puller (Sutter), and polished with MF-830 microforge (Narishige) to resistances of 3–5 MΩ. Hair bundles were deflected with a glass pipette mounted on a P-885 piezoelectric stack actuator (Physik Instrument). The tip of the pipette was fire-polished to ∼4 µm diameter to fit the shape of OHC bundles. The actuator was driven with voltage steps that were low-pass filtered at 10 KHz with a 900CT eight-pole Bessel filter (Frequency Devices). The output driving voltage to the actuator stack was monitored by an oscilloscope to ensure a rise time <50 µs. The tip of the probe was cleaned in chromic acid to allow adherence to hair bundles. Whole cell recordings were carried out and currents were sampled at 100 KHz with an EPC 10 USB patch-clamp amplifier operated by Patchmaster 2.35 software (HEKA). To record macroscopic currents, the patch pipette was filled with intracellular solution (140 mM KCl, 1 mM $MgCl_2$, 0.1 mM EGTA, 2 mM Mg-ATP, 0.3 mM Na-GTP and 10 mM H-HEPES, pH7.2). Cells were clamped at -70 mV.

## Data analysis

Data analysis was performed using Excel (Microsoft) and Igor pro 6 (WaveMetrics, Lake Oswego, OR). Calcium signal (△F/F) was calculated with the equation: $(F-F_0)/F_0$, where $F_0$ is the averaged fluorescence baseline at the beginning. Transduction current-displacement curves (I(X)) were fitted with a three-state Boltzmann model (*Grillet et al., 2009*). All data are mean ± SE. Student's two-tailed unpaired t test was used to determine statistical significance (*p<0.05, **p<0.01, ***p<0.001).

# Acknowledgements

We thank members of the laboratory for comments and criticisms. This work was supported by the NIH (UM; DC005965, DC007704, DC014713; BZ; DC015294), the Skaggs Institute for Chemical Biology (UM) and the Dorris Neuroscience Center (UM).

# Additional information

### Competing interests

UM: Founder of Decibel Therapeutics. The other authors declare that no competing interests exist.

### Funding

| Funder | Grant reference number | Author |
| --- | --- | --- |
| National Institute on Deafness and Other Communication Disorders | DC015294 | Bo Zhao |

| National Institute on Deafness and Other Communication Disorders | DC005965 | Ulrich Müller |
| National Institute on Deafness and Other Communication Disorders | DC007704 | Ulrich Müller |
| National Institute on Deafness and Other Communication Disorders | DC014713 | Ulrich Müller |

The funders had no role in study design, data collection and interpretation, or the decision to submit the work for publication.

## Author contributions

BZ, ZW, Conception and design, Acquisition of data, Analysis and interpretation of data, Drafting or revising the article; UM, Conception and design, Analysis and interpretation of data, Drafting or revising the article, Contributed unpublished essential data or reagents

## Author ORCIDs

Ulrich Müller, http://orcid.org/0000-0003-2736-6494

## Ethics

Animal experimentation: All of the animal experiments were approved by the ethics committee of the institutional animal care and use committees of The Scripps Research Institute (Approval number 080088).

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
