## [Decision Letter]

Thank you for submitting your work entitled "Super-resolution microscopy resolves protein organization in stereocilia critical for mechanosensory hair cell function" for consideration by *eLife*. Your article has been reviewed by 3 peer reviewers, and the evaluation has been overseen by Tanya Whitfield (Reviewing Editor) and Gary Westbrook as the Senior Editor. The reviewers have discussed the reviews with one another and the Reviewing Editor has drafted this decision to help you prepare a revised submission.

Summary:

This paper uses biochemical, imaging and genetic studies to characterise the protein *FAM65B* in sensory hair cells of the inner ear. Mutation in the *FAM65B* gene has previously been linked to deafness in humans. The current manuscript addresses the localisation and function of the *FAM65B* protein in sensory inner hair cells in the mouse. As you will see, all three reviewers have commended your work, but have also raised a number of concerns. These should be addressed as follows:

Essential revisions:

1) A thorough, objective and quantitative analysis of the STORM imaging data should be provided, together with appropriate statistical analysis where required, as requested by Reviewer 1. Inferences about direct interactions (between *FAM65B* and taperin, and *FAM65B* and RhoB) should be cautious unless further analysis can strengthen the evidence for these interactions.

2) Likewise, the interpretation of the co-IP results in HEK-293 cells should be discussed with appropriate caution, as requested by Reviewer 2. If additional co-IP results from a different cell line are available, these should be included.

3) A full structural analysis of the stereociliary bundle phenotype, including cohesion, growth, presence of tapers and stiffness should be provided, as requested by Reviewer 3.

A) It was agreed that further electrophysiological analyses of the transduction currents (calcium dependency, reverse polarity) would be beyond the scope of this manuscript, and so are not required. However, the authors should acknowledge and discuss any limitations of the methods they have used.

4) A comment on the presence or absence of the outer hair cells and of hair cell polarity defects in the mutant should be included.

5) Any discrepancies between the findings and those of Diaz-Horta et al. should be discussed, as requested by Reviewer 3.

6) The title should be revised, as requested by Reviewer 2, to reflect the findings of the manuscript more generally.

7) The name 'basilin' should be removed from the manuscript. There appears to be no strong reason to re-name the protein, and indeed, several arguments (as raised by Reviewer 3) the name 'basilin' is unsuitable. Retaining the name *FAM65B* will help with cross-reference to existing literature.

The full reviews are appended below for your information.

*Reviewer #1:*

This manuscript reveals the functional role of *FAM65B*, which is renamed as basilin, in structural integrity of stereocilia in hair cells. Most of the conclusions, based on genetic, biochemical and imaging data, are quite convincing. Nevertheless, the reviewer has several major concerns regarding the super-resolution data:

1) Figure 3 tries to show that HA-basilin in HEK cells forms string- or ring-like structures resembling those observed in stereocilia. However, the evidence was drawn from visual display of a small, selected region at the edge of one cell where the label density and the density of localization points is extremely low. In this case, the displayed ring-like structure in Figure 3-iii, which contains only 9 localization points, could simply be random coincidence. The string-like structures could also be artifacts resulted from colocalization with other cellular structures. The distance between some localization points and their nearest neighbors exceed 200 nm, which is an indication that these structures, if real, are unlikely maintained by basilin oligomerization alone. The manuscript would have to show statistically significant, objective analysis as to convincingly demonstrate the existence of large-size, ordered basilin structures in HEK cells. Otherwise, the biochemistry data can support the oligomerization capability of basilin, but cannot support that basilin rings are formed by basilin oligomerization.

2) Figure 6 uses the distinct morphology of basilin and taperin in STORM images to support that they have no direct interaction. However, judging from the two sets of single-color images, the basilin rings may still have certain degree of overlap with taperin disks. Therefore, these images by themselves cannot rule out the possibility that basilin is interacting with some of the taperins at the periphery of the taperin disks (maybe some post-translational modification?). The reviewer suggests more quantitative analysis of the STORM images: measuring the diameter of the two structures, and compare them to the diameter of the stereocilia measured in EM images. This comparison will be important to support the structural model proposed in Figure 8. (Optional request) If high quality two-color STORM imaging of basilin and taperin can be obtained, preferably in 3D, the conclusions can be further strengthened.

3) Figure 8 uses the colocalization of basilin-GFP and HA-RhoC in conventional fluorescence images to support their direct interaction. However, the basilin-taperin case in the same manuscript provides a perfect counter-example: colocalization in the conventional fluorescence images but no direct interaction and distinct morphology in STORM images. The reviewer thus suggests either showing STORM images of these two proteins, or changing the claims in the manuscript.

*Reviewer #2:*

The manuscript by Zhao et al. provides new insight into the role of Basilin in inner ear hair cell stereocilia morphogenesis and function While mutations in Fam65b (Basilin) had been shown to be responsible for deafness (Diaz-Horta et al., 2014), the role of this protein in hair cells has been unclear. Here the authors combine biochemical (mostly Co-IP), super-resolution microscopy and functional studies to provide insight into *FAM65B* (renamed "Basilin"), a protein previously shown to be linked to inherited forms of deafness but whose molecular and cellular function has remained unknown. The major findings are: 1) basilin is expressed at the taper end of the sensory inner hair cell stereocilia and hair bundles are disorganized in basilin deficient mice; 2) basilin forms head to head and tail to tail oligomers assembled in a ring like pattern at the base of each stereocilia; 3) oligomer formation depends on RhoC (a Rho kinase) binding which also regulates basilin activity. Interestingly, while basilin has been shown to interact with RhoA in neutrophils (Gao et al. 2015), the current work reveals only weak interaction between basilin and RhoA but highly efficient interaction with RhoC. Somewhat surprising, co-IP does not show any interaction between basilin and Cdc42 and Rac1, other Rho family members expressed in hair cells and demonstrated to play a role in cellular patterning, polarization and planar polarity (Grimsley-Myers et al., 2009; Kirjavainen et al. 2015). A major part of this report is based on Co-IPs (12 panels), which assess interaction between basilin and basilin constructs (to reveal interacting domains) with taperin, CLIC5, radixin, whirlin, RhoA, RhoC, Rac1 and Cdc42. Super-resolution microscopy demonstrates distinct localization between taperin and basilin (Figure 6) and linear or circular strings when overexpressed in HEK293 cells. The manuscript is well written and represents a compilation of a large amount of work including the generation of a basilin deficient mouse, immunolocalisation, judicious use of injectoporation with different constructs, co-Ip and Storm microscopy.

The title of the manuscript is not justified. While STORM microscopy resolves localization of basilin and taperin (Figure 6 and one panel in Figure 7) in stereocilia and HEK cells, this finding does not represent the entirety of the work. Rather a title that encompass the new findings i.e. the role of Basilin in hair cell stereocilia and its regulation by RhoC would be more appropriate.

Co-IP results in HEK-293 cells, in particular the lack of interactions between some of the players may not reflect the situation in native hair cells. Since a large portion of the work is based on co-IP, it is important to discuss these findings critically. Co-Ip using a different cell line such as PK1-CL4 cells or pull down assay might reveal different results. In particular, the lack of interaction with CLIC5, Radixin and whirlin is puzzling in part because Taperin has been shown to interact with CLIC5 (Salles et al. 2014). Have the authors also tested interactions with PTPRQ or Myosin VI also present in the tapering region of the stereocilia?

Since Rho family members have been shown to play a role in planar polarity, is there evidence of planar polarity defects (kinocilium position or bundle orientation) in the basilin deficient mouse?

*Reviewer #3:*

An interesting and exciting paper that utilizes an impressive suite of methodologies to provide evidence that *FAM65B*, thought by some to be a PX-BAR domain protein, (i) forms oligomeric ring-like structures in a Rho-C dependent at the base of hair-cell stereocilia and (ii) is required for the normal development and maintenance of hair-bundle structure in the mouse cochlea. The paper considerably extends upon the recent paper from Dias-Horta and colleagues showing that a deletion in *FAM65B* causes deafness in a human family and is required for the normal function and survival of hair cells in the zebrafish. Whilst the extent to which the paper reveals 'insights into the mechanisms by which proteins regulate hair cell function' remains debatable, this is an impressive body of work that certainly provides considerable insight into the ever increasing molecular complexity of the sensory hair bundle.

Additional comments:

Introduction: 'Mutation in nearly all of the genes…'. Are there mutations in genes that encode proteins that localise to the base of stereocilia that do not cause deafness?

The value of giving a protein yet another name is questionable. *FAM65B* appears to have quite a widespread distribution. It may be worth mentioning here that *FAM65B* may be a PX-BAR domain protein that could interact with lipids and regulate membrane curvature.

Aside from confusion causing by renaming, is basilin the best of names? It was originally used for a protein that was found in the basilar membrane but then dropped as the protein proved to be EMILIN-2. Many things are basally located in many structures, and then there are basal bodies, basal laminae…

Introduction: 'Basilin does not bind to taperin or any of its known binding partners'. Slightly ambiguous; rephrase.

Results first paragraph, first line: Is the ko mouse really a model for a human disease in which there is an in frame deletion in *FAM65B*?

Results, end of second paragraph: Are the OHCs even present, let alone functional? They disappear in the zebrafish MO knockdown so do they have hair bundles or survive to 4 weeks of age?

In the third paragraph of the subsection “Basilin expression in hair cells of the inner ear”: Is the commercially available antibody from Sigma any better than the two used by Dias -Horta et al.? Do the latter give staining in the Fam65b ko? Are these among the many antibodies that not of sufficient quality for immunolocalisation studies and/or STORM? Is there some explanation as to why Dias-Horta found localisation to the apical membrane and the membrane of stereocilia and this study does not? The 'validation' image in Figure 2 is remarkably black. Is the right panel gain and exposure matched to that on the left? There is considerable staining of the microvilli on the apical surface of all the supporting cells, and no images are shown of the apical surface of Hensens's cells. Do the microvilli on these cells have basal tapers, and are there any subtle defects in the structure or function of these microvilli? Is any lacz staining seen in Dieters' and pillar cells if the X-gal staining time is increased? Or is the microvillar staining seen on the surfaces of all supporting cells except Hensen's cells 'non-specific'?

In the first paragraph of the subsection “Basilin deficiency affects hair bundle morphogenesis and stereociliary growth”. Indicate in figure with arrows which stereocilia are abnormally thick and large. Are the stereocilia fusing?

In the second and third paragraphs of the subsection “Basilin deficiency affects hair bundle morphogenesis and stereociliary growth” and Figure 3 and Figure 4: Why or how does *FAM65B* affect hair bundle cohesion and growth? The TEM is not of very good quality. At best one can see the base of the tapered region in one stereocilium in the ko. Does the taper not form in the ko? This really needs to be addressed. Is taperin displaced up the stereocilia in the Fam65b ko? The apical protrusions are interesting; is this due to a loss of membrane cytoskeletal interactions at the base of the stereocilium? Are the perturbations seen really due to 'defects in bundle cohesion and growth'?

In the subsection “Defects in mechanotransduction in basilin-deficient hair cells”, transduction defects: Were the stiff probes used for the mutant hair cells fire polished to fit the distorted shape of the mutant hair bundles? The fluid jet is likely to be more suitable for direct measurements of transduction currents in the mutants. The cells in the wild type seem to have a rather broad operating range. Is the adaptation seen of the 'non-calcium dependent variety' described by Ricci and colleagues? Does it occur at positive holding potentials? What happened in the recordings where Ca^2+^ was reduced to 0.02 mM? Also, were reverse polarity currents measured (as indicated in methods?). Considering the localisation data and the theoretical contribution of the ankle region to hair-bundle properties it is surprising that the authors have not addressed whether the stiffness or resilience of the hair bundle is altered in any way in the ko. Displacement (Figure 5) is usually measured in micrometers not in micromoles.

In the last paragraph of the subsection “Defects in mechanotransduction in basilin-deficient hair cells” and Figure 5: How many cells were rescued by HA-basilin overexpression, and how many were not? How is taperin localised in rescued cells?

In the first paragraph of the subsection “Analysis of basilin and taperin distribution in hair cells” and Figure 6: The area boxed in Figure 6 is not that shown in Figure 6. It is surprising (in light of the aim of highlighting the power of STORM) that there is no dual labelling shown. The patterns of taperin and *FAM65B* are distinct but, judging from the images, most likely to overlap at some point. Whilst the antibodies to taperin and *FAM65B* would have to be directly conjugated to different fluorophores, a comparison with the distribution F-actin or a membrane marker should be possible; the latter might help confirm the membrane association of *FAM65B* that is depicted in the model shown in Figure 8, but is never explicitly tested or shown. Do the authors have a reason for excluding TRIOBP from the rootlet? Is it actually absent from this narrow region?

In the subsection “Basilin oligomerization is important for its function” and Figure 7: Two of the cells labelled with an asterisk are not obviously expressing N2-GFP, and one of these two has a disrupted hair bundle.

In the last paragraph of the subsection “RhoC binding to basilin is critical for basilin function in hair cells” and Figure 8: What are the morphological criteria for rescued and non-rescued hair cells?

Discussion, fourth paragraph: Are the ring like structures seen when *FAM65B* is expressed in heterologous cells associated with any structures? Surely physiological studies will be also be required to reveal if *FAM65B* oligomers form spring-link structures at the base of the stereocilia?

Methods: Minimal details are provided for STORM analysis.

[Editors' note: further revisions were requested prior to acceptance, as described below.]

Thank you for resubmitting your work entitled "Fam65B forms ring-like structures at the base of stereocilia critical for mechanosensory hair cell function" for further consideration at *eLife*. Your revised article has been favorably evaluated by Gary Westbrook (Senior editor), Tanya Whitfield (Reviewing editor), and three reviewers. The manuscript has been improved but there are some remaining issues that need to be addressed before acceptance, as outlined below:

1) There now appear to be three different versions of the revised title – one on the cover page, one on the first page of the manuscript and one in the rebuttal letter. Please clarify which of these you propose to use.

2) For the new measurements provided for the Fam65B rings, please state what the confidence bounds represent (SD, SEM or other).

3) Reviewer 1 recommends that the HEK images (Figure 8) should be removed from the manuscript. This was discussed among the reviewers, and on balance, it was agreed that removing these images was the best option. The reviewers found these observations very interesting, but thought that it was better to save the data for future follow-up work. Please therefore remove panel H from Figure 8 and adjust the legend etc. accordingly.

4) There was no mention of TRIOBP in the text, as stated in the rebuttal.

Further detail is provided in the assessments from Reviewers 1 and 3 below, for your information. Reviewer 2 was satisfied with your response and did not have any further comments.

Reviewer #1:

The revised manuscript has satisfactorily addressed previous concerns regarding Figure 7 (previous Figure 6, Fam65b-taperin interaction) and Figure 9 (previous Figure 8, basin-RhoC interaction). For Fam65b-taperin interaction, the measurement of their respective diameters supports the model that Fam65b may fill the entire space inside the taperin in. It is necessary, though, to describe in the methods section how the diameters are measured, and in the Results section whether the confidence bounds are standard deviations or S.E.M.

For the HEK cell STORM images in Figure 8, the reviewer thinks it is better to remove them. These images along do not support the oligomerization of Fam65b. If these structures were indeed Fam65b oligomers formed through the direct interaction between monomers, the localization points in the STORM images would have been much closer to each other. Instead, these structures could result from the interaction of Fam65b with other membrane or cytoskeleton structures in HEK cells. Of course, the observed sparse labeling may also be a consequence of low antibody staining efficiency, which is quite common. In either case, these STORM images do not strengthen the conclusions already supported by the biochemical analysis. The reviewer agrees with the authors that removing them does not affect the central conclusion of the manuscript.

*Reviewer #3:*

The authors have addressed most of the points raised by the reviewers well. The additional SEM data is fascinating, although it is hard to immediately understand why the thin, tapered region of the stereocilium should get longer in the absence of *FAM65B*. Whilst the quantitation of the diameters of the FAM65b and taperin staining patterns provide support for the structural model, these numbers are presumably derived from a subset of the FAM65b clusters seen at the base of the stereocilia; the standard errors (?) are small and judging from Figure 8 ring-shaped structures are very much in the minority. Some indication of the criteria used to select the images measured would be useful.

---

## [Author Response]

Essential revisions:

1) A thorough, objective and quantitative analysis of the STORM imaging data should be provided, together with appropriate statistical analysis where required, as requested by Reviewer 1. Inferences about direct interactions (between FAM65B and taperin, and FAM65B and RhoB) should be cautious unless further analysis can strengthen the evidence for these interactions.

We have quantified the diameter of the Fam65b circumferential ring and the taperin dense-core-like structure at the base of stereocilia. The diameter of Fam65b structure is 229 ± 7 nm (n=41) and taperin structure is 228 ± 9 nm (n=56). While the data indicate a potential overlap between the two structures (although this is difficult to say because labeling with the antibody adds to the dimension of the labeled structure), we provide substantial new data supporting the view that Fam65b and taperin do not interact directly. We performed Co-IP experiments in HEK293 cells (already shown in the original study) and in response to the reviewers' comments now also in CL4 cells, which have been used especially by the Bartles and Berryman laboratories to study interactions between proteins important for hair cell function. We provide positive controls demonstrating that we can reveal known interactions (e.g. taperin with CLIC5). In addition, we have carried out yeast-two-hybrid assays to evaluate interactions between Fam65b and taperin but cannot detect any interactions but we identify new interactions (e.g. Fam65b with RhoC). Taken together, the findings suggest that Fam65b does not bind directly to taperin. However, the two proteins might interact via an intermediate protein. We have revised the text to explain our data more clearly.

2) Likewise, the interpretation of the co-IP results in HEK-293 cells should be discussed with appropriate caution, as requested by Reviewer 2. If additional co-IP results from a different cell line are available, these should be included.

As outlined above, we have performed additional Co-IP experiments in CL4 cells and carried out additional yeast-two-hybrid experiments. We include appropriate positive controls. Our data confirm the published interaction of taperin with CLIC5 but do not reveal any interaction between taperin and Fam65b. The data are included in the revised manuscript.

3) A full structural analysis of the stereociliary bundle phenotype, including cohesion, growth, presence of tapers and stiffness should be provided, as requested by Reviewer 3.

We have substantially expanded our analysis of the morphological phenotype of hair cells using SEM and include an additional figure (Figure 4) as well as a supplementary figure (Figure 4—figure supplement 1). In addition, we have expanded Figure 3. We found that the morphological phenotype in hair cells is very heterogeneous with some cells showing defects in polarity, cohesion and/or growth already during early stages of hair cell development. In addition, the taper region of stereocilia appears abnormal and we observed unusual membranous protrusions from the apical cell surface. The defects at the taper are consistent with the localization of Fam65b to this region. We provide a quantitative analysis of the morphological phenotypes by grouping the hair cells according to different parameters into four morphological classes. We also provide a quantification of hair bundle polarity defects. Overall, the findings suggest that stereocilia grow and start to form a bundle with a staircase but tapers are abnormally shaped and bundles start to deteriorate before they reach maturity.

Given the strong structural defects and variability in morphology, we did not include measurements of hair bundle stiffness. Most biophysical measures on hair-bundle stiffness that have been published were collected from non-mammalian species that have substantially longer stereocilia than mice. These measurements are extremely difficult to quantify in murine hair cells and while we attempted measurements we could not determine reliable parameters that could be applied to the morphologically abnormal hair bundles in our mutant mice.

A) It was agreed that further electrophysiological analyses of the transduction currents (calcium dependency, reverse polarity) would be beyond the scope of this manuscript, and so are not required. However, the authors should acknowledge and discuss any limitations of the methods they have used.

We have revised the text to outline the limitations of our approach. The experiments are very interesting but we confront a similar problem as for stiffness measurements. It is very difficult to assign primary versus secondary effects due to heterogeneous degenerative changes in hair cells.

4) A comment on the presence or absence of the outer hair cells and of hair cell polarity defects in the mutant should be included.

We now include a quantification of hair cell survival at 2 and 4 weeks of age (Figure 7—figure supplement 1). There is minimal if any loss. We also have quantified hair bundle polarity at P2, when hair bundle morphology is sufficiently maintained to allow for a quantification using kinocilia as a landmark (Figure 3). We see a clear defect in polarity and include the data in the manuscript.

*5) Any discrepancies between the findings and those of Diaz-Horta et al. should be discussed, as requested by Reviewer 3.*

Unlike Diaz-Horta in studies with zebrafish, we did not observe significant hair cell loss at 2 and 4 weeks in mice, which suggests that the hearing phenotype in our rodent model is not caused by death of hair cells. There could be a species difference in Fam65b function and/or hair cells in zebrafish might die more readily when their hair bundles show morphological abnormalities. We discuss this in the revised manuscript.

Diaz-Horta et al. reported that Fam65b is localized in murine hair cells along the length of stereocilia. However, judging from the images in the manuscript, they show high expression at the base of stereocilia and less along their lengths. We purchased the two commercial antibodies used by Diaz-Horta and found that we could not detect convincing staining for Fam65b in hair cells. This might perhaps be explained by differences in antibody batches provided by the manufacturer at different times. We would also like to point out that the two commercial antibodies used by Diaz-Horta were raised against the N-terminus of Fam65b, which shows 70% similarity with Fam65a and Fam65c. We used an antibody to the C-terminus of Fam65b, which has very low homology with Fam65a/c. Perhaps hair cells express Fam65a, b, c and Diaz-Horta observed with their batch of antibody not only Fam65b. The antibody that we used is clearly specific as it does not stain hair cells from Fam65b mutant mice. We have revised the text to discuss the discrepancy between the studies and possible explanations for the difference.

*6) The title should be revised, as requested by Reviewer 2, to reflect the findings of the manuscript more generally.*

We have revised the title: "Fam65b and RhoC cooperate to form ring-like structures at the base of stereocilia critical for mechanosensory hair cell function"

7) The name 'basilin' should be removed from the manuscript. There appears to be no strong reason to re-name the protein, and indeed, several arguments (as raised by Reviewer 3) the name 'basilin' is unsuitable. Retaining the name FAM65B will help with cross-reference to existing literature.

We now refer throughout to Fam65b.

The full reviews are appended below for your information.

Reviewer #1:

This manuscript reveals the functional role of FAM65B, which is renamed as basilin, in structural integrity of stereocilia in hair cells. Most of the conclusions, based on genetic, biochemical and imaging data, are quite convincing. Nevertheless, the reviewer has several major concerns regarding the super-resolution data:

1) Figure 3 tries to show that HA-basilin in HEK cells forms string- or ring-like structures resembling those observed in stereocilia. However, the evidence was drawn from visual display of a small, selected region at the edge of one cell where the label density and the density of localization points is extremely low. In this case, the displayed ring-like structure in Figure 3-iii, which contains only 9 localization points, could simply be random coincidence. The string-like structures could also be artifacts resulted from colocalization with other cellular structures. The distance between some localization points and their nearest neighbors exceed 200 nm, which is an indication that these structures, if real, are unlikely maintained by basilin oligomerization alone. The manuscript would have to show statistically significant, objective analysis as to convincingly demonstrate the existence of large-size, ordered basilin structures in HEK cells. Otherwise, the biochemistry data can support the oligomerization capability of basilin, but cannot support that basilin rings are formed by basilin oligomerization.

We have revised the manuscript to address this point. As the reviewer correctly points out, we had to image regions at the edge of cells where label-density is low. The choice was deliberate because it is impossible to reach conclusions from areas with very dense labeling because there are just too many clustered dots, which cannot be assigned to different substructures. Also, we did not mean to imply that the HEK images show that Fam65b forms ring-like structures, just beads-on-a-string like arrangements that would be consistent with oligomerization (as observed in our biochemical analysis). The arrangements of the dots is not random, reproducibly seen in many cells and does not resemble any cellular structure that we have seen for any protein in HEK cells. It is difficult to quantify the data in an unbiased manner because we cannot resolve the structures in more densely labeled areas of a cell and are thus selective in the images that we show.

We have left the images in the revised manuscript and describe them much more carefully, also pointing out the limitations associated with the figure and finding. If the reviewer considers it is the better option, we are happy to remove the image from the manuscript. Not showing the data does not affect the central conclusion of our manuscript.

*2) Figure 6 uses the distinct morphology of basilin and taperin in STORM images to support that they have no direct interaction. However, judging from the two sets of single-color images, the basilin rings may still have certain degree of overlap with taperin disks. Therefore, these images by themselves cannot rule out the possibility that basilin is interacting with some of the taperins at the periphery of the taperin disks (maybe some post-translational modification?). The reviewer suggests more quantitative analysis of the STORM images: measuring the diameter of the two structures, and compare them to the diameter of the stereocilia measured in EM images. This comparison will be important to support the structural model proposed in Figure 8. (Optional request) If high quality two-color STORM imaging of basilin and taperin can be obtained, preferably in 3D, the conclusions can be further strengthened.*

We entirely agree with the reviewer that the STORM images cannot be used to rule in/out direct interactions between Fam65b and taperin. We formulated our text poorly. Our biochemical data (now extended), but not the imaging data, support that Fam65b and taperin do not interact directly.

Following the reviewer’s suggestion, we have now quantified the STORM images. The diameter of the Fam65b ring is 229 ± 7 nm (n=41) and of the taperin structure 228 ± 9 nm (n=56). While the data indicate a potential overlap between the two structures (although this is difficult to say because labeling with the antibody adds to the dimension of the labeled structure), we provide substantial new data supporting the view that Fam65b and taperin do not interact directly (new co-IPs from different cells; yeast-two-hybrid analysis). However, the similarity in diameter of the structures labeled by the two antibodies suggests that the two structures are potentially connected, perhaps by a linker protein. We discuss this now more carefully.

The antibodies we used for STORM were raised in rabbits. We tried several different antibodies and dyes, but we could not find a good combination that allowed us to carry out double-labeling for Fam65b and taperin. We are in the process of generating additional antibodies including monoclonals, but this will take substantial time.

*3) Figure 8 uses the colocalization of basilin-GFP and HA-RhoC in conventional fluorescence images to support their direct interaction. However, the basilin-taperin case in the same manuscript provides a perfect counter-example: colocalization in the conventional fluorescence images but no direct interaction and distinct morphology in STORM images. The reviewer thus suggests either showing STORM images of these two proteins, or changing the claims in the manuscript.*

We agree that co-localization does not mean that two proteins directly interact and we did not want to use the data as support for a direct interaction. To show that Fam65b and RhoC interact, we carried out yeast-two-hybrid analysis and Co-IP experiments. However, having identified an interaction between the two proteins in these heterologous systems, we wanted to confirm that they are present in hair cells in a similar distribution as a first step towards a validation that the biochemical data may be physiologically relevant. Clearly, the proteins are appropriately localized in hair cells to act in a common subcellular compartment. We attempted to further solidify the data and carry out STORM microscopy with RhoC antibodies but did not identify antibodies that worked with the technique. We have carefully modified the text to explain the localization data better. However, we still find that we provide significant evidence for a physical and functional interaction between Fam65b and RhoC: (i) Co-IP and yeast-two-hybrid to demonstrate direct interaction; (ii) demonstrating in cell lines that RhoC promotes Fam65b oligomerization; (iii) demonstrating in hair cells that Fam65b oligomerization and RhoC binding are critical for hair bundle morphogenesis.

Reviewer #2:

[…] The title of the manuscript is not justified. While STORM microscopy resolves localization of basilin and taperin (Figure 6 and one panel in Figure 7) in stereocilia and HEK cells, this finding does not represent the entirety of the work. Rather a title that encompass the new findings i.e. the role of Basilin in hair cell stereocilia and its regulation by RhoC would be more appropriate.

We have revised the title: "Fam65b and RhoC cooperate to form ring-like structures at the base of stereocilia critical for mechanosensory hair cell function".

*Co-IP results in HEK-293 cells, in particular the lack of interactions between some of the players may not reflect the situation in native hair cells. Since a large portion of the work is based on co-IP, it is important to discuss these findings critically. Co-Ip using a different cell line such as PK1-CL4 cells or pull down assay might reveal different results. In particular, the lack of interaction with CLIC5, Radixin and whirlin is puzzling in part because Taperin has been shown to interact with CLIC5 (Salles et al. 2014). Have the authors also tested interactions with PTPRQ or Myosin VI also present in the tapering region of the stereocilia?*

We have carried out additional experiments to strengthen the biochemical data. We have carried out immunoprecipitations in 293 and CL4 cells and we provide new yeast-two-hybrid data. The results from all experiments agree, revealing no interaction of Fam65b with taperin. As a positive control, we show that taperin interacts with CLIC5 as published. We also performed Co-IPs of the cytoplasmic domain of PTPRQ with Fam65b but did not observe an interaction. Finally, we carried out more yeast-two-hybrid screens but never identified interactions of Fam65b with proteins of the taperin/CLIC5/Whirlin complex, or with PTPRQ and MyoVI. We do not consider this a concern. The data suggest that there are several distinct protein complexes at the base, which is in agreement with the STORM images for Fam65b and taperin – clearly the proteins do not show the same distribution in hair cells. We anticipate that there might be a yet-to-be identified protein that links Fam65b to the taperin/CLIC5/Whirlin complex and we are trying to identify this protein. We also like to point out that there are other proteins such as TRIOBP that are abundant near the base of stereocilia and that form independent structures with no known biochemical link to the taperin/CLIC5/Whirlin complex.

We have revised the text and carefully address the limitations of our biochemical analysis.

Since Rho family members have been shown to play a role in planar polarity, is there evidence of planar polarity defects (kinocilium position or bundle orientation) in the basilin deficient mouse?

Yes, we have analyzed this. We now include data on hair-bundle polarity and observe clear polarity defects in hair cells at P2 (Figure 3).

Reviewer #3:

[…] Introduction: 'Mutation in nearly all of the genes…'. Are there mutations in genes that encode proteins that localise to the base of stereocilia that do not cause deafness?

We have corrected the text. So far all genes that have been localized to the base seem to be linked to deafness.

*The value of giving a protein yet another name is questionable. FAM65B appears to have quite a widespread distribution. It may be worth mentioning here that FAM65B may be a PX-BAR domain protein that could interact with lipids and regulate membrane curvature.*

We refer now to Fam65b throughout. We now discuss extensively that Fam65b might be a PX-BAR domain protein.

*Aside from confusion causing by renaming, is basilin the best of names? It was originally used for a protein that was found in the basilar membrane but then dropped as the protein proved to be EMILIN-2. Many things are basally located in many structures, and then there are basal bodies, basal laminae…* We refer throughout to Fam65b.

Introduction: 'Basilin does not bind to taperin or any of its known binding partners'. Slightly ambiguous; rephrase.

Sorry, that was not a good formulation. We have modified the text.

*Results first paragraph, first line: Is the ko mouse really a model for a human disease in which there is an in frame deletion in FAM65B?*

We have modified the text to be more accurate. We do not mimic the precise mutation linked to disease in humans and clearly describe this now in the text.

*Results, end of second paragraph: Are the OHCs even present, let alone functional? They disappear in the zebrafish MO knockdown so do they have hair bundles or survive to 4 weeks of age?*

We did not observe significant hair-cell loss at 2 and 4 weeks, which suggests that the hearing phenotype in our mice is not caused by death of hair cells. We have included the data in the revised manuscript. Diaz-Horta et al. studied zebrafish, and there could be a species difference in Fam65b function or hair cells; zebrafish hair cells might die more readily when their hair bundles show morphological abnormalities. We discuss the differences between the two studies in the revised manuscript. We also show data for mechanotransduction recordings, revealing functional defects in the hair cells.

In the third paragraph of the subsection “Basilin expression in hair cells of the inner ear”: Is the commercially available antibody from Sigma any better than the two used by Dias -Horta et al.? Do the latter give staining in the Fam65b ko? Are these among the many antibodies that not of sufficient quality for immunolocalisation studies and/or STORM? Is there some explanation as to why Dias-Horta found localisation to the apical membrane and the membrane of stereocilia and this study does not?

Diaz-Horta et al. reported that Fam65b is localized in murine hair cells along the length of stereocilia. However, judging from the images in the manuscript, they show very high expression at the base of stereocilia and less along their lengths. We purchased the two commercial antibodies used by Diaz-Horta and found that we could not detect convincing staining for Fam65b in hair cells. This might have to do with the fact that we likely obtained different batches of antibody from the manufacturer. We would also like to point out that the two commercial antibodies used by Diaz-Horta were raised against the N-terminus of Fam65b, which shows 70% similarity with Fam65a and Fam65c. We used an antibody to the C-terminus of Fam65b, which has very low homology with Fam65a/c. Perhaps hair cells express Fam65a, b, c and Diaz-Horta observed with their batch of antibody not only Fam65b. We have revised the text to discuss the discrepancy between the studies and possible explanations for the difference.

In our hands, the Sigma antibody works very well and does not stain hair cells from Fam65b mutant animals. We have purchased several batches and obtained reproducible results.

The 'validation' image in Figure 2 is remarkably black. Is the right panel gain and exposure matched to that on the left?

We apologize for confusing images. We now provide images with matched exposure.

There is considerable staining of the microvilli on the apical surface of all the supporting cells, and no images are shown of the apical surface of Hensens's cells. Do the microvilli on these cells have basal tapers, and are there any subtle defects in the structure or function of these microvilli? Is any lacz staining seen in Dieters' and pillar cells if the X-gal staining time is increased? Or is the microvillar staining seen on the surfaces of all supporting cells except Hensen's cells 'non-specific'?

Our analysis of hair bundle morphology suggests that Fam65b is not essential to form tapers but that stereocilia degenerate during their maturation. Thus, Fam65b could be present on other cell types even in microvilli without tapers.

Regarding the b-gal staining results and the immunohistochemistry, we have substantially revised the text and provide new data. The b-gal data show particular prominent expression in hair cells and Hensen's cell. However, b-gal is not a very sensitive read-out and we cannot exclude that there is weaker expression in other support cells. Using antibody staining, we see prominent labeling of hair cells but also clear signal in Hensen's cells and in some of the microvilli on the top of support cells. The staining is not seen in mutant mice, indicating that the signal is specific. We thus conclude that Fam65b is expressed in hair cells but also likely in support cells including Hensen's cells. New images for Hensen's cells are now included and we discuss the limitations of the b-gal staining data and our results more clearly.

By SEM, we have not been able to observe structural defect in microvilli of support cells and mention this now in the text.

In the first paragraph of the subsection “Basilin deficiency affects hair bundle morphogenesis and stereociliary growth”. Indicate in figure with arrows which stereocilia are abnormally thick and large. Are the stereocilia fusing?

We describe now in revised Figure 3 in detail structural defects in hair bundles. This is outlined in more detail below.

In the second and third paragraphs of the subsection “Basilin deficiency affects hair bundle morphogenesis and stereociliary growth” and Figure 3 and Figure 4: Why or how does FAM65B affect hair bundle cohesion and growth? The TEM is not of very good quality. At best one can see the base of the tapered region in one stereocilium in the ko. Does the taper not form in the ko? This really needs to be addressed. Is taperin displaced up the stereocilia in the Fam65b ko? The apical protrusions are interesting; is this due to a loss of membrane cytoskeletal interactions at the base of the stereocilium? Are the perturbations seen really due to 'defects in bundle cohesion and growth'?

We have substantially expanded our analysis of the morphological phenotype of hair cells using SEM. We found that the morphological phenotype in hair cells is very heterogeneous with some cells showing defects in polarity, cohesion and/or growth already during early stages of hair cell development. In addition, the taper region of stereocilia appears abnormal and we observed unusual membranous protrusions from the apical cell surface. The defects at the taper are consistent with the localization of Fam65b to this region. We provide a quantitative analysis of the morphological phenotypes by grouping the hair cells according to different parameters into four morphological classes. We also provide a quantification of hair bundle polarity defects. Overall, the findings suggest that stereocilia grow and start to form a bundle with a staircase but tapers are abnormally shaped and bundles start to deteriorate before they reach maturity.

We acknowledge that we do not fully understand the cellular mechanisms by which mutation in Fam65b causes defects in hair bundles but our data suggest that it has a critical structural role at the base of stereocilia. We discuss this carefully in the revised manuscript and try to avoid over-interpretation. However, even without a detailed mechanism, we still find that our findings are a significant step forward in our understanding of Fam65b function outlining subcellular distribution, physical and functional interactions with RhoC and a clear link between Fam65b and the morphogenesis of hair bundles.

In the subsection “Defects in mechanotransduction in basilin-deficient hair cells”, transduction defects: Were the stiff probes used for the mutant hair cells fire polished to fit the distorted shape of the mutant hair bundles? The fluid jet is likely to be more suitable for direct measurements of transduction currents in the mutants. The cells in the wild type seem to have a rather broad operating range. Is the adaptation seen of the 'non-calcium dependent variety' described by Ricci and colleagues? Does it occur at positive holding potentials? What happened in the recordings where Ca^2+^ was reduced to 0.02 mM? Also, were reverse polarity currents measured (as indicated in methods?). Considering the localisation data and the theoretical contribution of the ankle region to hair-bundle properties it is surprising that the authors have not addressed whether the stiffness or resilience of the hair bundle is altered in any way in the ko. Displacement (Figure 5) is usually measured in micrometers not in micromoles.

Given the strong structural defects and variability in morphology, we did not include measurements of hair bundle stiffness. Most biophysical measures on hair bundle stiffness that have been published were collected from non-mammalian species that have substantially longer stereocilia than mice. These measurements are extremely difficult to quantify in murine hair cells and while we attempted measurements we could not determine reliable parameters that could be applied to the morphologically abnormal hair bundles in our mutant mice.

Similarly, we are reluctant to characterize mechanotransduction currents in great detail in the genetically modified mice that we have available because of the severe defects in hair bundle morphology. Even if we can tease out some aspects relevant for adaptation or calcium dependence, we do not know the extent to which the results provide insights into Fam65b function or are a secondary consequence of the structural defects.

In the last paragraph of the subsection “Defects in mechanotransduction in basilin-deficient hair cells” and Figure 5: How many cells were rescued by HA-basilin overexpression, and how many were not? How is taperin localised in rescued cells?

We now provide a quantitative analysis of all rescue experiments shown in the manuscript. For rescue by HA-basilin expression we quantified two aspects of the phenotype, rescue of morphology and rescue of transduction. In Figure 5 we show the quantitative data for rescue of transduction, in Figure 8 we show the quantitative data for rescue of morphology. We have not attempted to localize taperin in the rescued cells and instead focused on the experiments suggested by the editorial summary. Should the reviewer insist, we can carry out the experiment.

In the first paragraph of the subsection “Analysis of basilin and taperin distribution in hair cells” and Figure 6: The area boxed in Figure 6 is not that shown in Figure 6. It is surprising (in light of the aim of highlighting the power of STORM) that there is no dual labelling shown. The patterns of taperin and FAM65B are distinct but, judging from the images, most likely to overlap at some point. Whilst the antibodies to taperin and FAM65B would have to be directly conjugated to different fluorophores, a comparison with the distribution F-actin or a membrane marker should be possible; the latter might help confirm the membrane association of FAM65B that is depicted in the model shown in Figure 8, but is never explicitly tested or shown. Do the authors have a reason for excluding TRIOBP from the rootlet? Is it actually absent from this narrow region?

We have revised the figures and text and now show matched images. Following the suggestions of several reviewers, we have now quantified the STORM images. The diameter of the Fam65b ring is 229 ± 7 nm (n=41) and of the taperin structure 228 ± 9 nm (n=56). The data indicate a potential overlap between the two structures (although this is difficult to say because laveling with the antibody adds to the dimension of the labeled structure). We discuss this now more carefully.

We tried extensively to use in STORM microscopy several different labels together with antibodies to Fam65b and taperin. We attempted to label membranes (Vybrant Dil, CM-DiI, DiO, biotin) as well as F-actin. However, we were unable to visualize double labeling with our instrumentation. For membrane staining, Vybrant DiI, Vybrant DiO, and Vybrant CM-DiI were readily taken up by supporting cells in the cochlea, but not very efficiently by the hair cells themselves; not just the stereocilia bundles, but also the apical membranes in hair cells did not stain significantly. Background was too high with biotin-based membrane staining methods. We were also looking for a good fluorophores coupled to phalloidin that could bleach with our laser illumination set-up (bleaching is critical for STORM), but using commercially available fluorophore-conjugated phalloidin we could not achieve good F-actin staining using STORM. We have more recently approached researchers at other institutions for collaborations to explore other technologies (SIM, STET) but do not yet have access to these instruments. Even without these data, we find that our studies make a substantial contribution and reveal clear differences in the distribution of taperin and Fam65b at the base of stereocilia. The diameter of the structures that we image is in good agreement with the diameter of stereocilia, suggesting localization of Fam65b near the membrane, as depicted in the diagram. Please note that we do not claim that our data show affinity of Fam65b for membranes or lipids (others have done that in the published literature). We merely depict a plausible model and have revised the text carefully to point this out.

We only excluded TRIOBP because it is more widely localized along rootlets and the figure is already crowded. However, we now mention TRIOBP in the text. It would indeed be very interesting to explore a connection between TRIOBP and Fam65b but we have not carried out experiments along these lines.

In the subsection “Basilin oligomerization is important for its function” and Figure 7: Two of the cells labelled with an asterisk are not obviously expressing N2-GFP, and one of these two has a disrupted hair bundle.

Expression in some of the displayed cells was low and did not display well in the provided image. We provide now a more suitable and representative images. We also show one cell that expresses very low levels of the protein and, as expected, no morphological disruption

*In the last paragraph of the subsection “RhoC binding to basilin is critical for basilin function in hair cells” and Figure 8: What are the morphological criteria for rescued and non-rescued hair cells?*

We specify this now in the text. The morphological defects of cultured Fam65b hair cells are more severe than in vivo. Most of them assume a round-shape or have strong defects in cohesion. This indicates that the bundles are likely fragile and that morphological defects are exaggerated by the dissection procedure. Following rescue by gene expression, the hair bundles show a clear organized morphology with a characteristic V-shape.

Discussion, fourth paragraph: Are the ring like structures seen when FAM65B is expressed in heterologous cells associated with any structures? Surely physiological studies will be also be required to reveal if FAM65B oligomers form spring-link structures at the base of the stereocilia?

As pointed out in response to the comments of reviewer 2, we have modified our section on the structures in heterologous cells – the data are qualitative in nature and if the reviewers consider them inappropriate for inclusion we can remove them without affecting the conclusions of the manuscript.

Regarding the "spring-like structures", we entirely agree that we do not have data to claim that the molecules form springs. We wanted to bring this up as an interesting discussion point. We have now further modified the text to express this explicitly: "One interesting possibility would be that basilin oligomers form spring-like structures near the membrane at the base of stereocilia that affect the mechanical properties of the bundle. While there is currently no experimental evidence for this model, we are carrying out structural studies to address this possibility."

Methods: Minimal details are provided for STORM analysis.

We now provide a detailed description of the method:

“Whole mount staining and super-resolution imaging

Cochlear whole mount staining was carried out as described (Zhao et al., 2014). Whole mount images were captured by fluorescent deconvolution microscopy (Deltavision). […] Additional reagents were: Alexa Fluor 488-phalloidin, Alexa Fluor 647-phalloidin (Invitrogen, Carlsbad, CA), Alexa Fluor 488 and Alexa Fluor 647 conjugated goat anti-rabbit secondary antibodies.”

[Editors' note: further revisions were requested prior to acceptance, as described below.]

Thank you for resubmitting your work entitled "Fam65B forms ring-like structures at the base of stereocilia critical for mechanosensory hair cell function" for further consideration at eLife. Your revised article has been favorably evaluated by Gary Westbrook (Senior editor), a Reviewing editor (Tanya Whitfield), and three reviewers. The manuscript has been improved but there are some remaining issues that need to be addressed before acceptance, as outlined below:

1) There now appear to be three different versions of the revised title – one on the cover page, one on the first page of the manuscript and one in the rebuttal letter. Please clarify which of these you propose to use.

We apologize for the confusion. We propose the following title: “Murine Fam65B forms ring-like structures at the base of stereocilia critical for mechanosensory hair cell function.” We have modified the text of the manuscript accordingly.

2) For the new measurements provided for the Fam65B rings, please state what the confidence bounds represent (SD, SEM or other).

The result is represented as Mean ± SEM. This is now stated in the first paragraph of the subsection “Analysis of Fam65b and taperin distribution in hair cells” (see also comments to reviewers below).

3) Reviewer 1 recommends that the HEK images (Figure 8) should be removed from the manuscript. This was discussed among the reviewers, and on balance, it was agreed that removing these images was the best option. The reviewers found these observations very interesting, but thought that it was better to save the data for future follow-up work. Please therefore remove panel H from Figure 8 and adjust the legend etc. accordingly.

The images have been removed and the text and figure legends have been adjusted.

4) There was no mention of TRIOBP in the text, as stated in the rebuttal.

We now cite TRIOBP: “In addition, TRIOPB forms rootlets that extend from stereocilia into the cuticular plate and traverse the taper region (Kitajiri et al. 2010).”

Further detail is provided in the assessments from Reviewers 1 and 3 below, for your information. Reviewer 2 was satisfied with your response and did not have any further comments.

Reviewer #1:

The revised manuscript has satisfactorily addressed previous concerns regarding Figure 7 (previous Figure 6, Fam65b-taperin interaction) and Figure 9 (previous Figure 8, basin-RhoC interaction). For Fam65b-taperin interaction, the measurement of their respective diameters supports the model that Fam65b may fill the entire space inside the taperin in. It is necessary, though, to describe in the methods section how the diameters are measured, and in the Results section whether the confidence bounds are standard deviations or S.E.M.

We have now revised the text accordingly: “We therefore carried out measurements to determine the diameter of Fam65b ring-like structures and taperin circles under in the tallest row of stereocilia of P7 inner hair cells using NIS-Elements AR analysis software. The ring-like Fam65b structure had a diameter of 229 ± 7 nm (mean ± SEM; n=41), while the taperin structure had a diameter of 228 ± 9 nm (mean ± SEM; n=56).”

Reviewer #3:

The authors have addressed most of the points raised by the reviewers well. The additional SEM data is fascinating, although it is hard to immediately understand why the thin, tapered region of the stereocilium should get longer in the absence of FAM65B. Whilst the quantitation of the diameters of the FAM65b and taperin staining patterns provide support for the structural model, these numbers are presumably derived from a subset of the FAM65b clusters seen at the base of the stereocilia; the standard errors (?) are small and judging from Figure 8 ring-shaped structures are very much in the minority. Some indication of the criteria used to select the images measured would be useful.

We have now revised the text accordingly: “We therefore carried out measurements to determine the diameter of Fam65b ring-like structures and taperin circles under in the tallest row of stereocilia of P7 inner hair cells using NIS-Elements AR analysis software. The ring-like Fam65b structure had a diameter of 229 ± 7 nm (mean ± SEM; n=41), while the taperin structure had a diameter of 228 ± 9 nm (mean ± SEM; n=56).”